# DNA methylation protects cancer cells against senescence

Xiaoying Chen [1,10], Kosuke Yamaguchi [1,2,10] ✉, Brianna Rodgers[1], Delphine Goehrig[3], David Vindrieux[3], Xavier Lahaye [4], Matthieu Nolot [1], Laure Ferry[1], Sophie Lanciano[5], Nadine Martin[3], Pierre Dubus[6,7], Fumihito Miura[8], Takashi Ito [9], Gael Cristofari [5], Nicolas Manel [4], Masato T. Kanemaki [2], David Bernard [3] & Pierre-Antoine Defossez [1] ✉

Inhibitors of DNA methylation such as 5-aza-deoxycytidine are widely used in experimental and clinical settings. However, their mechanism of action is such that DNA damage inevitably co-occurs with loss of DNA methylation, making it challenging to discern their respective effects. Here we deconvolute the effects of decreased DNA methylation and DNA damage on cancer cells, by using degron alleles of key DNA methylation regulators. We report that cancer cells with decreased DNA methylation—but no DNA damage—enter cellular senescence, with G1 arrest, SASP expression, and SA-β-gal positivity. This senescence is independent of p53 and Rb, but involves p21, which is cytoplasmic and inhibits apoptosis, and cGAS, playing a STING-independent role in the nucleus. Xenograft experiments show that tumor cells can be made senescent in vivo by decreasing DNA methylation. These findings reveal the intrinsic effects of loss of DNA methylation in cancer cells and have practical implications for future therapeutic approaches.

DNA methylation is an abundant modification in mammals[1]. In differentiated murine or human cells, about 80% of cytosines in the CpG sequence context are marked with a methyl group on carbon 5, or CpG methylation. CpG methylation is epigenetic as it is transmitted from one cell to its descendants, and it influences genome activity, without changing the sequence itself.

Different cell types have different methylomes, owing to a dynamic equilibrium between the deposition and removal of DNA methylation[2]. Removal can occur by active mechanisms[3], or passively, by failure of the DNA methylation maintenance machinery to reproduce the mark following DNA replication.

DNA methylation maintenance involves two key enzymes[4]. The first is the DNA methyltransferase DNMT1, which can act on hemimethylated DNA[5]. The activity of DNMT1 is regulated by intramolecular inhibition, which limits its activity on unmethylated DNA[6]. During and after replication[7,8], this activity is enabled thanks to another essential actor: ubiquitin-like PHD and RING finger domain-containing protein UHRF1[9,10]. This protein interacts with the DNA replication machinery[11,12] and binds hemimethylated DNA; it can then mono-ubiquitinate histones and the PCNA-associated protein PAF15; these modified proteins, in turn, bind an auto-inhibitory region in DNMT1 and release its activity[13–15].

[1]Université Paris Cité, CNRS, Epigenetics and Cell Fate, Paris, France. [2]Department of Chromosome Science, National Institute of Genetics, Research Organization of Information and Systems (ROIS), Mishima, Shizuoka, Japan. [3]Equipe Labellisee La Ligue Contre Le Cancer, Centre de Recherche en Cancerologie de Lyon, Inserm U1052, CNRS UMR 5286, Centre Leon Berard, Université de Lyon, Lyon, France. [4]Institut Curie, PSL Research University, Paris, France. [5]University Cote d'Azur, INSERM, CNRS, Institute for Research on Cancer and Aging of Nice (IRCAN), Nice, France. [6]Department of Tumor Biology, Centre Hospitalier Universitaire de Bordeaux, Bordeaux, France. [7]BRIC U1312, INSERM, Bordeaux Institute of Oncology, Université de Bordeaux, Bordeaux, France. [8]Life Science Data Research Center, Graduate School of Frontier Sciences, the University of Tokyo, Chiba, Japan. [9]Department of Biochemistry, Kyushu University Graduate School of Medical Sciences, Fukuoka, Fukuoka, Japan. [10]These authors contributed equally: Xiaoying Chen, Kosuke Yamaguchi. ✉e-mail: yamako0801@icloud.com; pierre-antoine.defossez@cnrs.fr

Functionally, DNA methylation is linked to gene expression. The DNA methylation of gene bodies correlates positively with their expression level. In contrast, high levels of DNA methylation at CpG-rich promoters cause transcriptional repression. In healthy somatic cells, this transcriptional repression by DNA methylation applies to imprinted genes, certain tissue-specific genes, including germline genes, and to repeated elements[2,16,17].

The pattern of DNA methylation in cancer cells is different from that of WT cells, combining losses of DNA methylation over large domains called "Partially Methylated Regions", and foci of hyper-methylation, in particular over CpG island promoters, including those of repressed Tumor Suppressor Genes[18,19]. These premises provided the rationale for epigenetic therapy in cancer, with the hypothesis that decreasing DNA methylation in cancer cells may normalize their tumor suppressor gene expression and/or exacerbate their DNA methylation anomalies past a tolerable level[20].

One approach to diminishing DNA methylation in cancer cells has proven particularly successful and led to the first FDA-approved epigenetic drug for cancer treatment, Vidaza[21]. This approach relies on a cytosine analog, 5-aza-deoxy-cytidine (5-aza-dC), being incorporated into replicating DNA. Active DNMT1 reacting with 5-aza-dC leads to the formation of a covalent adduct, which is then removed by the DNA repair machinery. In the process, the DNMT1 molecules are destroyed, leading to passive DNA demethylation.

5-aza-dC, or its precursor 5-aza-C, has been extremely useful molecular probe to examine the phenotypic consequences of decreasing DNA methylation in cancer cells. Depending on cellular context, dose, and duration of treatment, these consequences in most cases are cell differentiation or cell death[21], with senescence reported in a few rare occurrences[22–25]. A key phenotype with consequences for cancer biology and treatment is that 5-aza-dC treatment induces the reactivation of repeated elements, producing cytoplasmic nucleic acids that trigger an interferon response[26,27].

The numerous studies carried out with 5-aza-C and 5-aza-dC have moved the field of cancer epigenetics forward, yet they also suffer from a serious caveat: 5-aza-dC causes loss of DNA methylation but also DNA damage, and one cannot occur without the other. Therefore, it is challenging to deconvolute the effects of DNA demethylation from those of DNA damage, which itself can induce apoptosis or senescence.

One way to disentangle these effects is to perform loss-of-function studies on proteins that maintain DNA methylation, such as DNMT1 or UHRF1, yet existing methods all have limitations. Constitutive genetic knock-out is not applicable to essential genes and may select for adaptation. Conditional knock-out is hampered by delayed kinetics, while RNAi and shRNA often fail to achieve total mRNA depletion, and their effects depend on the protein turnover rate. These various shortcomings can be circumvented by using protein degron approaches[28].

We recently generated degron alleles of DNMT1 and UHRF1 in colorectal cancer cells[29]. The rapid, complete, and synchronous protein degradation permitted by this approach allowed us to delineate molecular mechanisms by which UHRF1 maintains DNA methylation homeostasis[29]. In the current study, we have used these degron tools to answer a key question in cancer epigenetics: "What are the consequences to a cancer cell when DNA methylation levels decrease in the absence of DNA damage?". There is some evidence to suggest that decreasing DNMT1 or UHRF1 in non-transformed cells can direct cells towards senescence[30–32], but whether this is also true in cancer cells, and the nature of the mechanisms involved, are unclear at present.

We report that cells with lowered DNA methylation −but no DNA damage− go into senescence, with the typical attributes of G1 accumulation, enlarged nuclei, a Senescence-Associated Secretory Phenotype (SASP), and positivity for Senescence-Associated Beta-Galactosidase (SA-β-gal). Mechanistically, this senescence is induced more rapidly by inhibiting UHRF1 than DNMT1. It is independent of p53 and the p16/Rb pathways. Instead, it involves cytoplasmic p21, which

protects the senescent cells from apoptosis, and cGAS, acting in the nucleus independently of STING. We observed DNA-demethylation-induced senescence in multiple cancer lines originating from different organs, and demonstrated with xenografts that it also occurs in vivo.

These findings reveal the intrinsic effects of loss of DNA demethylation in cancer cells and have practical implications for future therapeutic approaches.

## Results

### Prolonged depletion of DNMT1 and/or UHRF1 triggers senescence in colorectal cancer cell lines

To examine the consequences of prolonged DNA demethylation, we used the colorectal cancer cell line HCT116, exploiting the auxin-inducible degron (AID1) system[33]. As described in our previous publication[29], these HCT116 cells (WT, U$^{AID1}$, D$^{AID1}$ and UD$^{AID1}$) express the plant E3 ligase OsTIR1 under a doxycycline (Dox)-inducible promoter, and were edited so that UHRF1, DNMT1, or both, had a mini-AID (mAID) tag, allowing for the proteins of interest to be rapidly degraded in the presence of auxin (Fig. 1A).

To characterize the effect of chronic depletion of UHRF1, DNMT1, or both, we treated the HCT116 WT and its derivatives with Dox/Auxin continuously for 8 days. A time-course western blot proved that UHRF1 and/or DNMT1 were completely degraded throughout the duration of the experiment (Fig. 1B). We also collected DNA samples for Whole-Genome Bisulfite Sequencing (WGBS) every 2 days and verified that the rate of DNA methylation loss was smaller in DNMT1 degron cells, greater in UHRF1 degron cells, and greatest in UHRF1/DNMT1 degron cells (Source Data), matching the results we previously published[29]. Lastly, we quantified cell proliferation: the cells depleted of DNMT1 displayed a significant growth defect compared to the WT; this defect was worse in UHRF1-depleted cells, and worst in cells lacking both DNMT1 and UHRF1 (Fig. 1C).

To further explore this proliferation defect, we next performed a cell-cycle analysis. HCT116 WT, U$^{AID1}$, D$^{AID1}$, and UD$^{AID1}$ cells were collected at 0, 4, or 8 days of Dox/Auxin treatment, and stained with bromodeoxyuridine (BrdU) and propidium iodide (PI) to profile the cell-cycle proportions. Through analysis with flow cytometry, we observed a significant decrease in the population of cells in the S phase of the cell cycle after UHRF1 or DNMT1 depletion compared to the WT (Fig. 1D), which parallels the loss of proliferation seen in the cell growth curve. We also noted a significant accumulation of the cell population in the G1 phase after UHRF1 and/or DNMT1 removal (Fig. 1D).

The reduced proliferation of cells lacking UHRF1 and/or DNMT1 could involve increased apoptosis and/or increased senescence, so we went on to examine both phenotypes. We began by an Annexin V analysis, which detects apoptotic cells. When comparing day 0 to day 8 of auxin treatment by this assay, we detected no extensive increase in apoptosis (Supplementary Fig. 1A). For further confidence in excluding apoptosis, we performed a TUNEL assay followed by fluorescent microscopy to observe potential DNA fragmentation as seen in apoptotic cells. After 8 days of auxin treatment, the cells depleted for UHRF1 and/or DNMT1 did not display any detectable staining for fragmented DNA (Supplementary Fig. 1B), unlike the positive control (cells treated with DNase) (Supplementary Fig. 1C). These data support the notion that the loss of cell growth upon UHRF1 and/or DNMT1 depletion is not caused by increased cell death, leaving increased senescence as another possible explanation.

Senescent cells display a number of morphological alterations, including enlarged nuclei[34]. After quantification of the DAPI signal in the various experimental conditions, we determined that indeed the loss of UHRF1 and DNMT1 resulted in increased nuclear size (Fig. 1E, F), further suggesting a senescent phenotype in the degron cells.

To further confirm this possible phenotype, we next assayed senescence-associated-beta-galactosidase (SA-β-gal), a common marker of the increased lysosomal activity present in senescent cells[35]. This

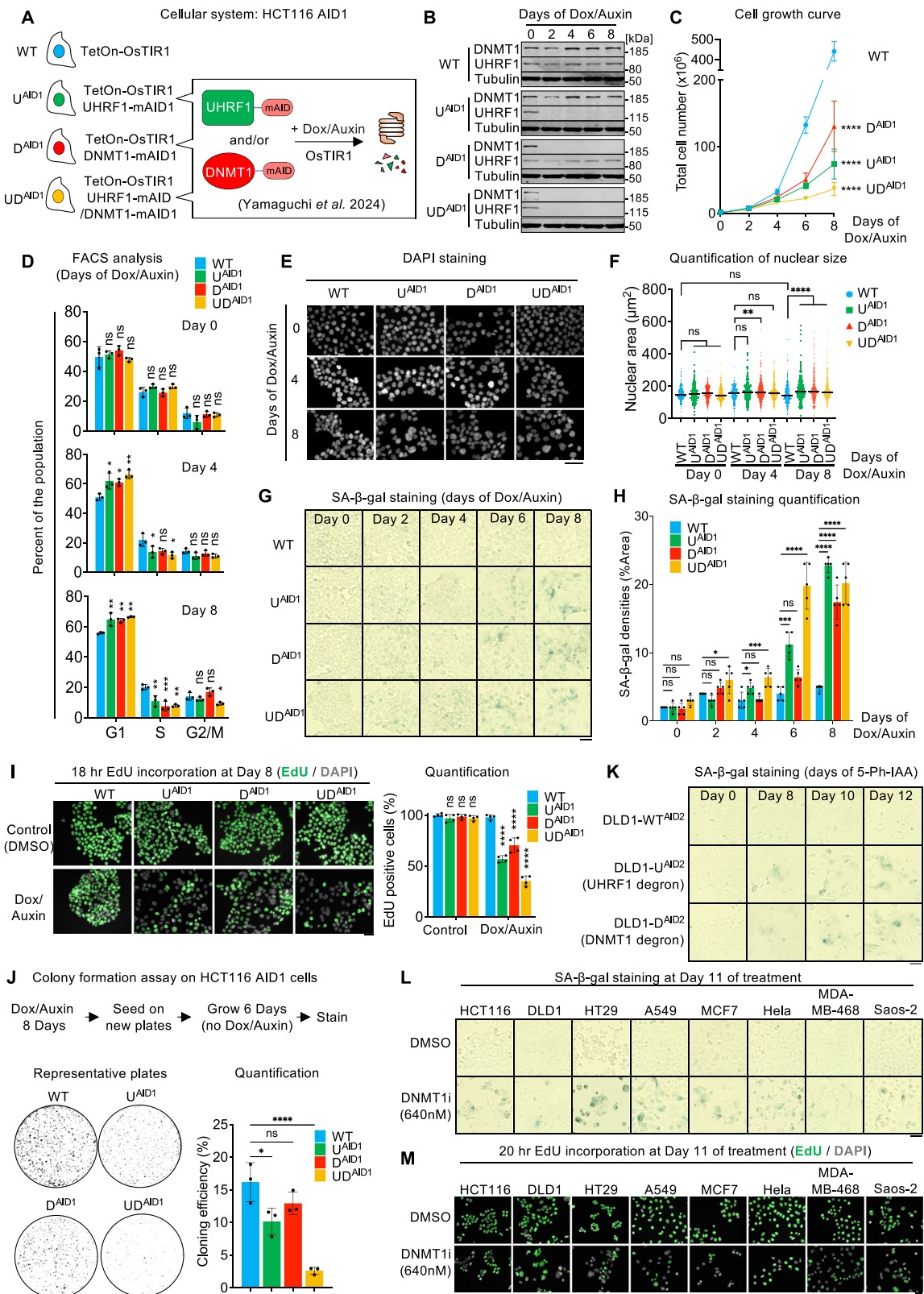

revealed a significant increase of SA-β-gal positive cells after the depletion of DNMT1, which became more pronounced with time. The increase was stronger in cells lacking UHRF1, and strongest in cells lacking both DNMT1 and UHRF1 (Fig. 1G, H), which parallels their degree of growth impairment.

We performed 2 additional experiments to ascertain the phenotype of the degron cells. The first experiment aimed at testing if the

degron cells were still progressing through the cell cycle, albeit at a slower pace. For this, after treating the cells with Dox/Auxin for 8 days, we subjected them to an 18-h long exposure to EdU. This duration was long enough to label 100% of the DMSO-treated cells; in contrast, about half of the cells lacking UHRF1, DNMT1, or both failed to incorporate any EdU over the 18-h period (Fig. 1I), which rules out a simple lengthening of the cell cycle. This assay also reveals a heterogeneous

**Fig. 1 | Prolonged UHRF1 or DNMT1 depletion triggers senescence in cancer cells. A** The auxin degron system in HCT116 cells conditionally expressing OsTIR1. **B** Western blot validating the degradation of UHRF1 and/or DNMT1 upon Dox/Auxin treatment. **C** Cell growth curve. *N* = 3 biological replicates. **D** Cell-cycle analysis by FACS with BrdU and PI staining. *N* = 3 biological replicates. **E** Representative images of DAPI staining in the indicated lines. **F** Nuclear area determination by DAPI staining. *N* = 441, 441, 393, 583, 326, 300, 459, 500, 325, 326, 339 and 356 nuclei, respectively. **G** Representative images of SA-β-gal staining after Dox/Auxin treatment. **H** Quantification of SA-β-gal staining. *N* = 5 fields of view. **I** Visualization and quantification of EdU incorporation by the indicated cells during an 18-h pulse. *N* = 4 fields of view. **J** Colony-forming assay on the indicated degron cells. N = 3 biological replicates. **K** Representative images of SA-β-gal staining after 5-Ph-IAA (1 μM) treatment with the indicated DLD1 degron cells constitutively expressing OsTIR1 (F74G). **L** Representative images of SA-β-gal staining after treatment of the indicated cells with DMSO or DNMT1 inhibitor GSK-3685032 (640 nM). **M** Visualization of EdU incorporation by the indicated cells during a 20-h pulse. All scale bars are 50 μm. Data of **C**, **D**, **H**–**J** are presented as mean ± SD and analyzed by one-way ANOVA test with Dunnett's multiple comparisons test. Data of **F** are presented as median and analyzed by Kruskal–Wallis test with Dunn's multiple comparisons test. In all figures, we use the following convention: *$p < 0.05$, **$p < 0.01$, ***$p < 0.001$, ****$p < 0.0001$, ns non-significant. Source data are provided as a Source Data file.

response, as some cells do maintain EdU incorporation even after UHRF1 and/or DNMT1 depletion, in agreement with our FACS data (Fig. 1D).

The second experiment tested whether the growth impairment experienced by the cells was reversible. For this, after treating the cells with Dox/Auxin, we seeded them in new plates and allowed them to form colonies in the absence of treatment. Cells that had undergone depletion of UHRF1, DNMT1, or both, formed colonies that were smaller and less numerous (Fig. 1J). This effect was more marked for the UHRF1-depleted cells than for DNMT1-depleted cells, and was particularly strong for UHRF1/DNMT1-depleted cells, with a decrease of colony formation greater than 6-fold (Fig. 1J). The effect was not fully penetrant, suggesting that not all cells reach an irreversible growth arrest after 8 days of removing UHRF1, DNMT1, or both.

These experiments show that HCT116 cells lacking DNMT1, UHRF1, or both present 4 phenotypes found in senescent cells: accumulation in G1 phase of the cell cycle; enlargement of the nucleus; expression of SA-β-gal; and irreversible growth arrest. The prevalence of these phenotypes increases with duration of depletion, and matches the degree of DNA methylation loss we described in our previous paper[29]: cells lacking UHRF1 lose DNA methylation more rapidly than cells lacking DNMT1, and present stronger senescent-like phenotypes. Cells deprived of both UHRF1 and DNMT1 present the steepest loss of DNA methylation and also the strongest senescence-like phenotypes.

### Senescence in the depleted cells is linked to loss of DNA methylation and occurs in multiple cellular backgrounds

The loss of either DNMT1 or UHRF1 leads to senescence in HCT116 cells, suggesting that this response may be caused by decreased DNA methylation. To test this possibility, we rescued the degron cells with previously characterized UHRF1 and DNMT1 variants[29], which do, or do not, support DNA methylation (Supplementary Fig. 1D–I). We then performed SA-β-gal staining in the rescued cells after 8 days of auxin treatment. We observed a strong correlation between loss of DNA methylation and senescence: the cells in which DNA methylation was sustained (WT or TTD mutant UHRF1, WT or PBD mutant DNMT1) were negative for SA-β-gal staining, while the cells in which DNA methylation was decreased (empty vectors, UBL or PHD or SRA or RING mutant UHRF1, UIM or H3 binding or catalytic mutant DNMT1) stained positively for SA-β-gal staining (Supplementary Fig. 1E, F, H, I). These data suggest that loss of DNA methylation underpins the senescence of HCT116 degron cells in our study.

Next, we sought to determine whether the observations made in HCT116 could be extended to other cellular backgrounds. In our previous publication[29], we generated auxin-inducible degron alleles of UHRF1 and DNMT1 in colorectal DLD1 cells, using the AID2 system in which the OsTIR1 has the F74G modification and the activator is 5-Ph-IAA[36]. We treated the DLD1 cells with 5-Ph-IAA for 8 days or longer, then performed SA-β-gal staining: this revealed an increase of SA-β-gal positivity after the removal of either UHRF1 or DNMT1 (Fig. 1K), extending the link between DNA demethylation and senescence beyond HCT116 cells.

We next asked whether the senescence phenotype seen with the removal of UHRF1 or DNMT1 was specific to the degron system or if we could confirm it with an independent intervention. For this we treated various cancer cell lines (HCT116, DLD1, HT29 (all three colorectal adenocarcinoma), A549 (non-small-cell lung adenocarcinoma), MCF7 (breast adenocarcinoma), HeLa (cervical carcinoma), MDA-MD-468 (breast adenocarcinoma), and Saos-2 (osteosarcoma)) with a DNMT1-selective noncovalent inhibitor that does not cause DNA damage, GSK3685032[37]. We incubated the cells with 640 nM inhibitor for 11 days, then performed the SA-β-gal assay (Fig. 1L). This inhibition of DNMT1 led to an increase of SA-β-gal positivity in all cell lines, along with a markedly decrease capacity to incorporate EdU during a 20-h pulse (Fig. 1M and Supplementary Fig. 1J) further validating the hypothesis that DNA demethylation induced by UHRF1 or DNMT1 removal triggers senescence, a phenomenon consistent in multiple cancer cell types as well as with various methods of depletion.

### Bulk transcriptional profiling of degron cells reveals an interferon response and a SASP signature

Having identified this senescence induced by decreased DNA methylation, we moved on to a detailed molecular characterization by performing an RNA-seq timed series (0, 2, 4, 6, and 8 days of Dox/Auxin treatment, Fig. 2A). These experiments extend the limited transcriptome analysis we previously reported at days 0 and 4[29], yielding important insight as reported below.

Principal component analysis plots showed that biological triplicates of each condition were highly similar (Supplementary Fig. 2A). Untreated WT and degron lines also display very comparable RNA-seq patterns, meaning that the AID tag has no obvious effect on cells, as we previously reported[29]. The first principal component explains 65% of the variance, and along this first axis, the WT samples at all days are tightly clustered. In contrast, the DNMT1-AID samples move progressively farther away from the WT samples with increasing time. The UHRF1-AID samples move away from WT faster and farther than DNMT1-AID, while the UHRF1/DNMT1-AID samples move the fastest and the farthest. The second principal component explains 18% of the variance, and the 6-day treated WT and 8-day treated WT cells segregate from untreated, 2-day, or 4-day treated WT along this axis (Supplementary Fig. 2A), which likely reflects the known effects of auxin treatment[36]. To remove the influence of auxin, in all subsequent analyses, we compared the Dox/Auxin-treated degron lines to the WT counterpart that had been Dox/Auxin-treated for the same duration.

We first computed differentially expressed genes (DEGs) upon UHRF1 and/or DNMT1 depletion over time (Fig. 2B). For all degron cells, the number of DEGs increased steadily with time, and there were more upregulated than downregulated genes. The DNMT1 degron cells showed the lowest number of DEGs at all time points, while UHRF1 degron cells had a higher number of DEGs, and the UHRF1/DNMT1 degron cells had the highest number, which parallels both the degree of DNA methylation loss[29] and the prevalence of senescent cells (see previous section). There is a high overlap of genes induced by removing UHRF1, DNMT1, or both (Fig. 2C), consistent with the

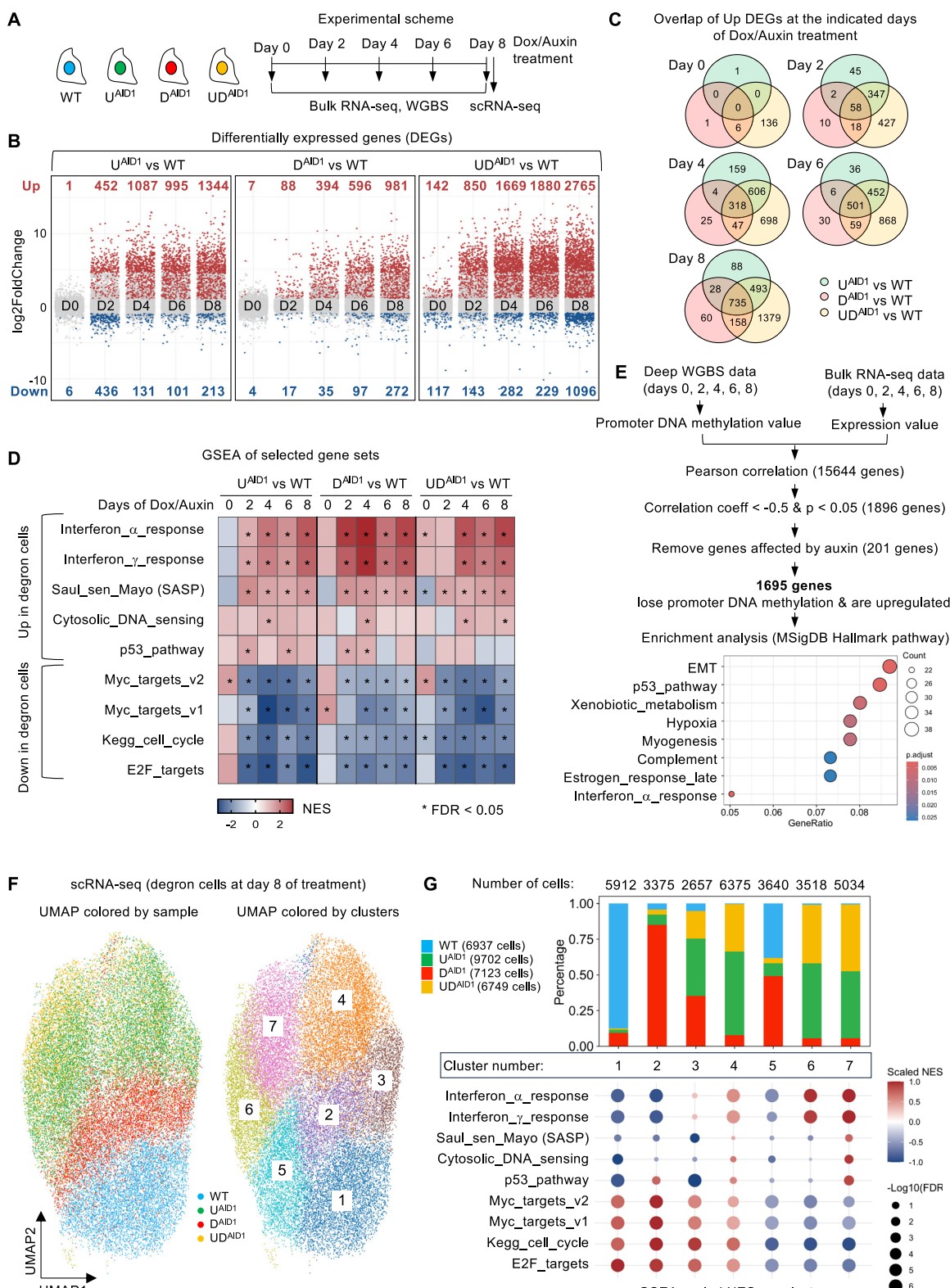

**A** Experimental scheme

**B** Differentially expressed genes (DEGs)

**C** Overlap of Up DEGs at the indicated days of Dox/Auxin treatment

**D** GSEA of selected gene sets

**E** Deep WGBS data (days 0, 2, 4, 6, 8) / Bulk RNA-seq data (days 0, 2, 4, 6, 8)

**F** scRNA-seq (degron cells at day 8 of treatment)

**G**

possibility that they are responding to the loss of DNA methylation caused by removing either protein.

We next performed gene set enrichment analysis (GSEA) on our data; a non-exhaustive list of gene sets is shown in Fig. 2D. Several signatures were downregulated after Dox/Auxin treatment: those are generally related to cell cycle (E2F targets, Myc targets, cell cycle), which agrees with the decreased cellular proliferation we

observed upon treatment of the degron cells. Conversely, many signatures were upregulated in treated cells, and among them were interferon (IFN) response signatures. This aligns with other studies where the interferon response is induced when DNA methylation decreases, after treatment with the DNA demethylating agent such as 5-aza-C or 5-aza-dC[26,27], or following RNA interference on DNMT1 or UHRF1[38,39].

**Fig. 2 | Senescence caused by loss of DNA methylation is accompanied by an interferon response and a SASP signature. A** Experimental scheme of sample collection for bulk RNA sequencing (RNA-seq), whole-genome bisulfite sequencing (WGBS) and single cell RNA sequencing (scRNA-seq). **B** Number of differentially expressed genes (DEGs) at the indicated days of Dox/Auxin treatment. DEGs were identified using DESeq2 tool. P values were calculated using a two-sided Wald test and adjusted for multiple comparisons via the Benjamini-Hochberg (BH) method. Up-regulated genes (red): adjusted *p* value < 0.05 and log2 FoldChange > 1. Down-regulated genes (blue): adjusted *p* value < 0.05 and log2 FoldChange < −1. Light gray dots: no significant change. **C** Venn diagram of significantly up-regulated DEGs. **D** Gene set enrichment analysis (GSEA) of selected gene sets from the Molecular Signatures Database (MSigDB). The heatmap represents the normalized enrichment score (NES) in each comparison at the indicated days. Significantly enriched gene sets are shown in red; significantly depleted gene sets are displayed

in blue. Conditions with an asterisk are significant with a false discovery rate (FDR) < 0.05. **E** Combined analysis of the WGBS and bulk RNA-seq data identifies genes for which loss of promoter DNA methylation correlates with gene activation. The promoter region is defined from −1200 bp to 300 bp relative to the transcription start site (TSS). Enrichment analysis was performed using clusterProfiler package. Significantly enriched terms were identified by a one-sided Fisher's exact test. Enriched Hallmark pathways (MSigDB) for these genes are shown with dot plot indicating gene ratio and BH-adjusted FDR. **F** UMAP visualization of scRNA-seq data from degron cells at day 8 of treatment. Left: colored by sample; right: colored by clusters. **G** Analysis of the corresponding Leiden clusters. Top: bar plot showing the number and proportion of cells per cluster from each sample. Bottom: GSEA analysis of selected gene sets per cluster. Dot size represents significance (−log10 FDR), and color indicates scaled NES. Source data are provided as a Source Data file.

---

We then sought to more accurately identify the molecular actors involved in the interferon response by a thorough examination of the RNA-seq dataset. The mRNAs encoding IFN lambda (*IFNL1*, *IFNL2*) were induced from day 4 of the time course. To validate these results, we performed RT-qPCR on degron cells at days 0 and 8 of treatment. This experiment confirmed the increase of *IFNL1* and *IFNL2* (Supplementary Fig. 2B), and demonstrated the induction of *IFI27* and *ISG15*, two genes stimulated by interferon. Furthermore, we collected serum-free conditioned medium (CM) from 8-day Dox/Auxin-treated WT/AID lines and, for a positive control, from PolyI:C-transfected cells (Supplementary Fig. 2B). This serum-free CM was utilized in ELISA experiments, which revealed that the treated degron HCT116 cells secrete IFN lambda 1 (Supplementary Fig. 2C).

Our RNA-seq data showed that HCT116 cells express receptors for IFN lambda (*IFNLR1* and *IL10RB*, Supplementary Data 1), raising the possibility that the treated degron cells could trigger a paracrine reaction in cells treated with CM. To test this idea, we supplemented the CM from 8-day auxin-treated WT/AID lines, and from cells made senescent by the DNA-damaging agent etoposide with 10% FBS (Supplementary Fig. 2D). Subsequently, we cultured HCT116 WT cells with this CM for 24 h, extracted total RNA from these cells, and performed RT-qPCR analysis. We found that conditioned medium from cells depleted of UHRF1 and/or DNMT1 induced interferon-responsive genes (*IRF7*, *IFI27*, *ISG15*, *OAS1*, *OAS1L*) in WT cells, whereas CM from etoposide-treated cells did not (Supplementary Fig. 2E). These data further substantiate the interferon response of the degron cells.

Our GSEA analysis (Fig. 2D) used a SASP signature containing 125 genes[40]. We refined the SASP signature specific to our system by examining the expression of individual SASP genes in the RNA-seq data (Supplementary Fig. 2F). All three degron lines displayed a similar SASP pattern: they induced IL32 and other cytokines, CCL5 and other chemokines, MMP1, MMP13, and other proteases, IGFBP3, IGFBP4 and other growth factors, and ICAM1, TNFRSF1B, and other receptors. These findings from RNA-seq analysis were confirmed by RT-qPCR (Supplementary Fig. 2G). This SASP signature was induced fastest and strongest in the UD^AID1 degron cells, more slowly and less strongly in the U^AID1 degron cells, and was more delayed and subdued in the D^AID1 degron cells (Supplementary Fig. 2F), which again parallels the kinetics of DNA demethylation and senescent phenotypes. The SASP phenotype is not restricted to HCT116 degron cells, as we also observed the induction of the same SASP genes in DLD1 degron cells upon prolonged UHRF1 or DNMT1 depletion (Supplementary Fig. 2H).

**A joint WGBS/bulk RNA-seq analysis to identify direct consequences of promoter demethylation**
Some promoters are hypermethylated in HCT116 cells and lose DNA methylation upon removal of UHRF1 and/or DNMT1—this could explain some or all of the transcriptional inductions seen in the bulk RNA-seq data. To ascertain this, we conducted a joint analysis of the WGBS and bulk RNA-seq data (Fig. 2E). This revealed that some

genes in the EMT signature or p53 pathway could be activated as a result of promoter DNA demethylation. In contrast, the Interferon or SASP responses are unlikely to be mere consequences of DNA methylation loss over previously repressed promoters.

**Expression of repeated elements and global chromatin status**
A commonly observed consequence of treating cells with DNA methylation inhibitors is the upregulation of certain transposable elements (TEs) and other repetitive DNA sequences. To assess whether that took place in the degron cells, reanalyzed our bulk RNA-seq data with the TEtranscripts package[41]. We observed a dynamic pattern of TE expression that varied between cells (Supplementary Fig. 2I). U^AID1 degron cells expressed a constant low number of TEs (less than twenty), but their nature changed over the course of treatment. The induction of TE expression in D^AID1 degron cells was gradual and culminated at ~30 TEs expressed by day 8. Lastly, the UD^AID1 degron cells expressed the most TEs (~80 by day 8). In all cells, we saw a range of elements expressed, belonging to the SVA, Simple repeat, DNA, SINE, LINE, and LTR families.

Loss of heterochromatin has been shown to be implicated in replicative senescence[42,43], so we also examined the level of various histone marks in the treated cells. We performed this analysis at day 4 of treatment, to observe effects that are more likely causes, and less likely consequences, of senescence. In addition, doing the experiment at day 4 decreases the confounding effect of increased nuclear size. We observed no loss of H3K9me2 or H3K27me3 in the degron cells (Supplementary Fig. 2J). Regarding H3K9me3, there was no loss in U^AID1 or D^AID1 cells, which is in line with the limited re-expression of repeated elements noted above, and a mild decrease in the UD^AID1 cells, which also correlates with a greater induction of repeated elements. Another salient observation was the marked increase of H3K9ac in U^AID1 and UD^AID1 cells (Supplementary Fig. 2J).

**Single-cell RNA-seq deconvolutes the transcriptional signatures**
To evaluate the heterogeneity of the transcriptional responses and to deconvolute the expression signatures observed, we performed single-cell RNA-seq on degron cells treated with Dox/Auxin for 8 days. A first UMAP analysis showed that the cells cluster by type: WT and DNMT1-degron cells are well segregated, whereas UHRF1-degron and UHRF1/DNMT1-degron cells are interspersed in the UMAP plot (Fig. 2F). We next refined the analysis by identifying cell clusters with the Leiden algorithm (Fig. 2F), and observed that a solution with 7 cell clusters provided meaningful data (Fig. 2G).

Clusters 1 to 4 contain cells with an active proliferation signature. Cluster 1 contains cells that are mostly WT and have no interferon, p53, or SASP signature. Cluster 2 contains cells that are mainly DNMT1-degron and have no interferon or SASP signature, but a weak p53 activation signature. Cluster 3 contains the 3 types of degron cells, but very few WT. These cells have no p53 or SASP activation, but a weak interferon signature. Finally, cluster 4 contains cells that are mostly

U^AID1 or UD^AID1 degron. They have the weakest proliferation signature, a stronger interferon response, p53 activation, and a weak SASP activation.

Clusters 5 to 7 contain cells with an inactive proliferation signature. Cluster 5 contains WT and D^AID1 degron cells. These cells display no p53 or SASP activation and no interferon response. These cells are likely arrested but not senescent. Clusters 6 and 7 are mostly made up of U^AID1 and UD^AID1 degron cells. Both clusters show a prominent activation of the interferon response. In addition, cluster 7 shows activation of the p53, SASP signatures, and cytosolic DNA sensing signaling.

Based on the number and type of cells per cluster, it can be estimated that the percentage of non-proliferating cells is about 17% for WT, 33% for D^AID1 degron cells, 50% for U^AID1 degron cells, and 66% for UD^AID1 degron cells. These numbers are highly consistent with the proliferation data of Fig. 1.

In addition, the data show that the p53 response can occur without an interferon response (Cluster 2), and that the interferon response can occur without a p53 response (Clusters 3 and 6), suggesting they are independently regulated. The SASP only occurs in cells that have activated cytosolic DNA sensing pathway and p53 pathway (Cluster 7). We next sought to identify the molecular determinants of these responses.

## DNA demethylation-induced senescence is not accompanied by DNA damage and is independent of p53 and p16/Rb

One of the established triggers of cellular senescence is DNA damage, which can act by activating p53[44]. To understand whether the senescence observed in the UHRF1/DNMT1-depleted HCT116 cells may involve DNA damage, we first carried out immunofluorescence on degron cells with an antibody for γ-H2AX, a marker of DNA double-strand breaks (Fig. 3A). As a positive control, we treated cells with hydroxyurea (HU), which interferes with DNA replication; this gave rise to a clear γ-H2AX signal, as expected. In contrast, even upon prolonged Dox/Auxin treatment (8 days), the loss of UHRF1 and/or DNMT1 did not lead to any detectable increase of the γ-H2AX signal (Fig. 3A and Supplementary Fig. 3A). The γ-H2AX signal examined by immunofluorescence in cells at the intermediate 4-day timepoint was also below the detection threshold (Supplementary Fig. 3A).

To confirm these data, we used western blotting, including three positive controls. Two controls involved an acute 4-h treatment with HU or with etoposide, which inhibit topoisomerase II, thus causing double-strand breaks. The last control involved cells with a persistent DNA-damage response (DDR). For this, we treated cells with etoposide for 24 h, then switched to fresh medium. The treatment drove the cells into DNA damage-induced senescence, as attested by positive SA-β-gal staining (Source Data). We tested 4 different actors of the DNA damage response: γ-H2AX, Chk1, Chk2, and p53. All four proteins behaved as anticipated in the positive controls, with increases of γ-H2AX, phospho-Chk1, phospho-Chk2, and p53 (Fig. 3B). In contrast, none of these four markers were induced in the auxin-treated degron cells relative to the auxin-treated WT cells (Fig. 3B). Altogether these data suggest that DNA damage is unlikely to be a major driver of senescence in the degron cells.

To further delineate the senescence mechanisms active in our cells, we next investigated the p16/Rb pathway. p16/INK4a inhibits Cyclin D/CDK complexes, thus preventing the phosphorylation of Rb and entry into S phase, and the activation of the p16/Rb pathway is frequently observed in senescent cells[45]. We first analyzed the expression of CDKN2A gene, which encodes p16 in HCT116 cells depleted for UHRF1 and/or DNMT1 using RT-qPCR, and found that CDKN2A was significantly upregulated in the U^AID1 and UD^AID1 cells after 8 days of Dox/Auxin treatment (Supplementary Fig. 3B). Western blotting confirmed that p16 was induced at the protein level in these samples (Supplementary Fig. 3C). In parallel, we inspected Rb expression and phosphorylation, and found both Rb and phospho-Rb protein levels decreased upon the removal of either UHRF1 or UHRF1

and DNMT1 (Supplementary Fig. 3D). This suggests that the p16/Rb pathway may be involved in driving cell cycle arrest and senescence in our system.

To test this possibility, we applied a shRNA approach to deplete p16. The WT and degron cells were infected with a lentiviral vector expressing shp16, or the negative control shLUC, then selected to ensure permanent shRNA expression. After these steps, the cells were treated with Dox/Auxin and we assessed their growth, SA-β-gal activity, and gene expression pattern by RT-qPCR (Fig. 3C). The knockdown of p16 did not ameliorate the slow proliferation of degron cells (Fig. 3D), nor their increased SA-β-gal positivity (Fig. 3E). RT-qPCR confirmed sustained suppression of the p16 gene following an 8-day Dox/Auxin treatment, yet the knockdown did not mitigate SASP gene expression (Fig. 3F). Collectively, these results indicate that p16 is not a key enforcer of cell cycle arrest or senescence in cells lacking UHRF1 and/or DNMT1.

To further test the contribution of p53 and Rb to senescence in our system, we exploited the oncogenic proteins of human papillomavirus HPV16: E6, which degrades p53, and E7, which inactivates Rb. The WT and degron cells were infected with an empty lentiviral vector or with a vector expressing E6 and E7, and selected to ensure stable expression (Fig. 3G). We verified that the p53 protein was degraded upon E6 and E7 expression (Supplementary Fig. 3E). As expected, the inactivation of p53 and Rb caused WT cells to grow faster (Fig. 3H); in contrast this treatment did not rescue the proliferation defect of D^AID1 or UD^AID1 lines treated with auxin (Fig. 3H). The treatment ameliorated the growth of U^AID1 cells treated with auxin, yet they remained severely growth-impaired (Fig. 3H). Moreover, the inactivation of p53 and Rb failed to reduce the SA-β-gal positivity (Fig. 3I) or SASP gene expression in the treated degron cells (Fig. 3J).

To further test this point, we treated HCT116 p53^−/− cells with the DNMT1 inhibitor; they also become SA-β-gal positive (Supplementary Fig. 3F). This result fits with the observations of Fig. 1L, showing senescence induced by loss of DNA methylation in cells lacking functional p53 (DLD1 (S241F mutation), HT29 (R273H mutation), HeLa (inactivation by E6), MDA-MB-468 (R273H mutation), Saos-2 (deletion)) and/or Rb (HeLa (inactivation by E7), MDA-MB-468 (deletion), Saos-2 (deletion)). The lack of p53 and/or Rb in these cells was verified by western blotting (Supplementary Fig. 3G).

Taken together, these results establish that the senescence observed in HCT116 after prolonged DNA demethylation is independent of the p53 and p16/Rb pathways.

## L1 transposable elements do not drive senescence caused by loss of DNA methylation

Given the known role of L1 in replicative senescence[46], we examined L1 expression more specifically, at the protein level, using an antibody against ORF1p. This revealed an induction of L1 ORF1p in UHRF1/DNMT1 degron after 8 days of depletion (Supplementary Fig. 3H). To determine if L1 expression is causal for senescence, we performed shRNA against L1, using a previously published vector[47]. We infected and selected the degron cells, then treated them with 5-aza-C to induce ORF1p to verify the efficacy of the shRNAs (Supplementary Fig. 3I). Next, we asked whether stably knocking down L1s had an effect on proliferation or senescence in our cells. As shown in Supplementary Fig. 3J, the shORF1 vector did not rescue the growth defect in any of our cell lines. Besides the growth phenotype, we also assessed whether shORF1 remediated SA-β-gal staining or SASP expression, and found that it did not (Supplementary Fig. 3K, L).

From these results, we conclude that L1 elements are not necessary for senescence in our system.

## The senescent cells are not aneuploid

Aneuploidy has been proposed as a key trigger into senescence[48,49], so we examined whether this phenomenon might be at play in the senescence observed upon loss of DNA methylation.

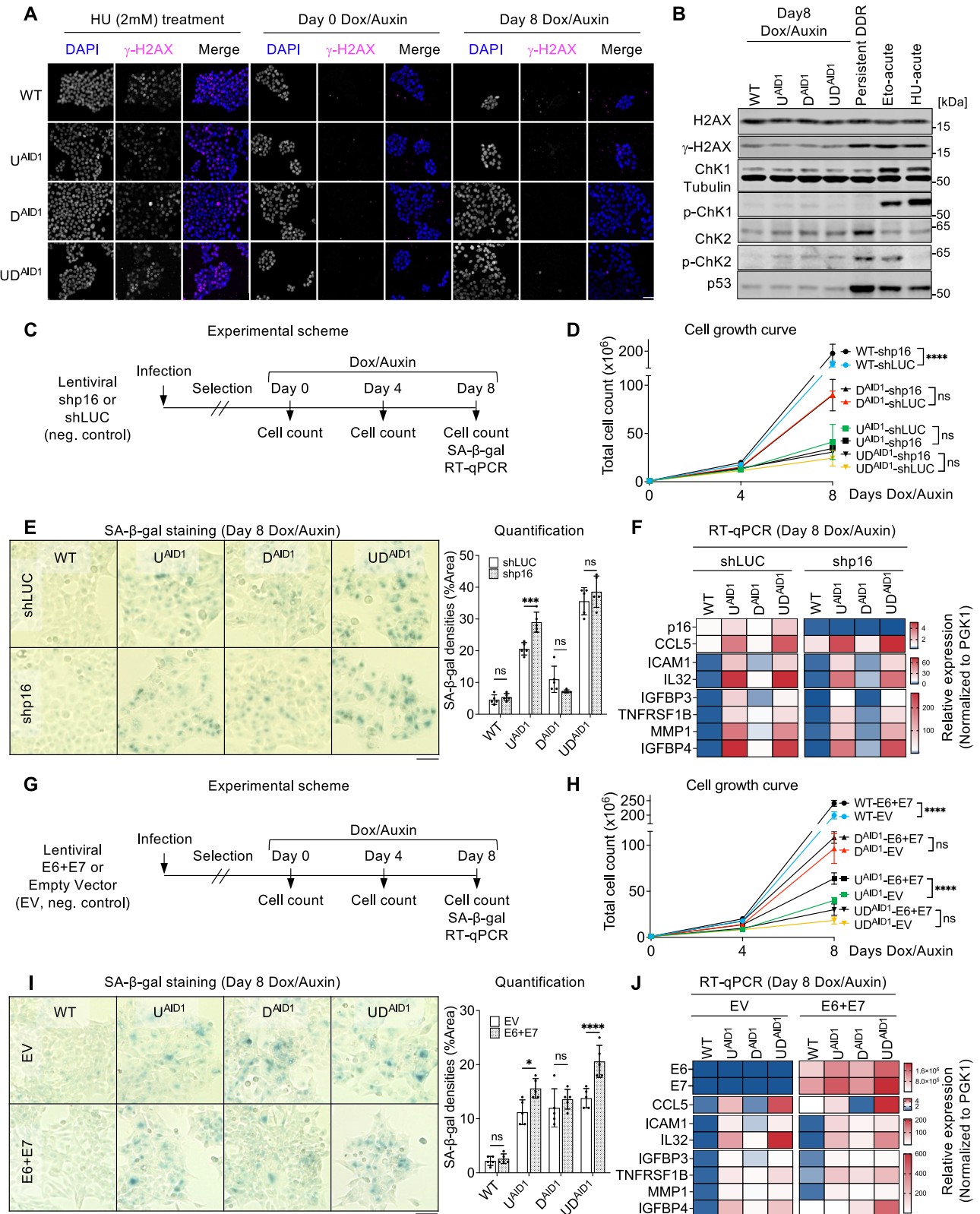

For this, we leveraged the single-cell RNA-seq dataset mentioned above and inferred copy number variations (CNVs) from the expression data with the InferCNV algorithm, validating our analysis by using the reference data of aneuploid cells provided in the package (Supplementary Fig. 3M). We observe in our HCT116 cells the local gains and losses of genomic DNA that have been reported for HCT116 cells by gDNA sequencing[50]. Furthermore, the analysis shows no difference

between the degron and control HCT116 cells, arguing against the implication of aneuploidy in our system.

**p21 is found in the cytoplasm of cells with decreased DNA methylation and contributes to their resistance against apoptosis**
Having excluded the role of canonical pathways (p53, p16/Rb) in the senescence of cells chronically deprived of UHRF1 and/or DNMT1, we

**Fig. 3 | Senescence caused by loss of DNA methylation is independent of p53 and p16/Rb. A** γ-H2AX immunofluorescence in HCT116 lines upon hydroxyurea treatment (HU, 2 mM, 4 h) or upon Dox/Auxin treatment. **B** Immunoblots for total H2AX, γ-H2AX, Chk1, phospho-Chk1 (Ser317), Chk2, phospho-Chk2 (Thr68), and p53 after 8-day Dox/Auxin treatment. HCT116 cells were used as positive controls: Persistent DNA damage response (DDR): Etoposide (10 μM, 24 h), then plain medium for 5 days; Eto-acute: Etoposide, 10 μM, 4 h; HU-acute: hydroxyurea, 2 mM, 4 h. **C** Scheme of p16 depletion experiments. **D** Total cell numbers at the indicated days of Dox/Auxin treatment. $N = 3$ technical replicates. **E** SA-β-gal staining and quantification of indicated lines. $N = 5$ fields of view. **F** RT-qPCR results of p16 (*CDKN2A*)

and selected SASP genes. **G** Scheme of E6 + E7 expression experiments. **H** Total cell numbers at the indicated days of Dox/Auxin treatment. $N = 3$ technical replicates. **I** SA-β-gal staining and quantification of indicated lines. $N = 5$ fields of view. **J** RT-qPCR results of *E6*, *E7*, and selected SASP genes. All scale bars are 50 μm. Data of **D**, **E**, **H**, **I** are presented as mean ± SD and analyzed by two-way ANOVA with Sidak's multiple comparisons test. Heatmap data of **F**, **J** are presented as mean from three technical replicates. In all figures we use the following convention: \**p < 0.05, \*\**p < 0.001, \*\*\**p < 0.0001, ns non-significant. Source data are provided as a Source Data file.

investigated the potential role of other factors. We first exploited our RNA-seq data and observed that, similarly to *CDKN2A*/p16, *CDKN1A*/p21 was also elevated at various time points in all the degron lines, and we confirmed this observation by RT-qPCR (Fig. 4A). p21 was also induced in the degron lines after 8-day Dox/Auxin treatment at protein level (Fig. 4B), indicating a potential role in the senescence process.

To test this possible role, we generated a lentiviral construct expressing an shRNA targeting p21, stably expressed it in degron cells, then quantified growth and markers of senescence (Fig. 4C). We first monitored cell proliferation after p21 depletion and found that the absence of p21 significantly increased the cell growth of WT cells, as expected, but that it decreased the numbers of U$^{AID1}$ and UD$^{AID1}$ lines (Fig. 4D), indicative of slower proliferation and/or increased apoptosis in these cells.

p21 has diverse functions depending on its cellular localization[51]: nuclear p21 inhibits various Cyclin/CDK complexes and restricts cell cycle progression, whereas cytoplasmic p21 has pro-survival effects by restraining JNK signaling and caspase activity[52]. Cell cycle exit and resistance to apoptosis are both common phenotypes of senescent cells[53], therefore, we next assessed the subcellular location of p21 in the degron cells. For this, we separated the nuclear and cytoplasmic fractions of HCT116 cells after 0 and 8-day Dox/Auxin treatment and performed western blotting. Upon validating the depletion of UHRF1 and DNMT1, the results revealed that the majority of the p21 protein present in senescent cells resided in the cytoplasm (Fig. 4E and Supplementary Fig. 4A). We then measured the apoptosis marker and DNA damage marker after p21 removal in degron cells, and observed increased abundance of cleaved poly ADP-Ribose Polymerase (PARP), as well as increased γ-H2AX after 8-day Dox/Auxin treatment (Fig. 4F), which was not observed in the same conditions at 0-day Dox/Auxin treatment (Supplementary Fig. 4B). Both results are compatible with an anti-apoptotic role of p21 in the cells rendered senescent by lack of DNMT1 and/or UHRF1. To further ascertain this possible role, we directly measured the level of apoptosis in our cells by an Annexin V assay (Fig. 4G). Compared to the control conditions, the loss of p21 led to an increase of apoptosis in the conditions depleted for UHRF1 and/or DNMT1 (Fig. 4H). These data again support the notion that the senescent cells resist apoptosis in part via p21. Additionally, *p21* knockdown led to a partial reduction of SA-β-gal positivity (Fig. 4I), but RT-qPCR showed that *p21* knockdown did not decrease SASP expression (Fig. 4J). Collectively, these data show that p21 is located in the cytoplasm of senescing cells with lowered DNA methylation and promotes their survival, but is not involved in the SASP induction.

Lastly, we investigated the possible causes of p21 upregulation in cells depleted of UHRF1 and/or DNMT1. The *CDKN1A* gene, which encodes the p21 protein, contains a large CpG island overlapping with the gene promoter, suggesting it could be subject to regulation by DNA methylation. To examine this possibility, we exploited our WGBS data; this revealed that the *CDKN1A* locus was mostly unmethylated in all conditions (Supplementary Fig. 4C). Furthermore, a combined analysis of RNA-seq and WGBS data showed that *CDKN1A* transcriptional induction bore no correlation to level of promoter DNA methylation (Supplementary Fig. 4D). These two results suggest that the induction of p21 is not caused by loss of DNA methylation.

We then investigated candidate transcriptional regulators that could explain the p21 induction in senescent cells. The first actor we considered was c-Myc, which is known to repress *CDKN1A* transcription[54–56]. Three lines of evidence suggest that c-Myc activity decreases in the degron cells as they become senescent upon exposure to Dox/Auxin: first, the c-Myc signatures are downregulated in the GSEA analysis (see Fig. 2D above); second, the c-Myc activity—as reflected in the VIPER (Virtual Inference of Protein-activity by Enriched Regulon analysis) score[57]—also decreases (Supplementary Fig. 4E); third, the levels of c-Myc protein (Supplementary Fig. 4F) diminish. Therefore, the decrease of c-Myc may account, at least in part, for the induction of p21. Another negative regulator of *CDKN1A* transcription is TFAP4 (Transcription Factor AP4)[58,59]. Similarly to c-Myc, we saw that the VIPER score of TFAP4 decreases in the degron cells treated with auxin (Supplementary Fig. 4E), concomitant with a decrease abundance of TFAP4 protein (Supplementary Fig. 4F). TFAP4 has been shown in other contexts to prevent senescence[60], so it may play this role in our system as well. These results on the transcriptional control of p21 are summarized in Supplementary Fig. 4G.

## Nuclear cGAS contributes to the SASP response and lysosomal activity of senescent cells

The nuclear acid sensor pathway cGAS/STING, comprised of cyclic GMP-AMP synthase (cGAS), stimulator of interferon genes (STING), and downstream signaling adaptors, can be involved in cellular senescence and SASP-derived inflammation[61–63], prompting us to examine its possible implication in our system.

To this end, we first performed RT-qPCR on the HCT116 cells at various timepoints after Dox/Auxin treatment to measure the expression of cGAS. We observed that the loss of UHRF1 and DNMT1 together triggered a dramatic upregulation of *cGAS*; this was less prominent after UHRF1 depletion and not significant after DNMT1 loss (Fig. 5A). We further validated this with western blot (Fig. 5B), and found that the cGAS protein was undetectable in untreated HCT116 WT cells, in agreement with previous work[64]. In contrast, the depletion of UHRF1 and DNMT1, or to a lesser extent the depletion of UHRF1 alone, led to an induction of cGAS protein that was visible as early as day 2 of treatment, and increased continuously over time (Fig. 5B). The *cGAS* gene contains a CpG island that was highly methylated in the initial conditions (Supplementary Fig. 5A), and the expression of *cGAS* in treated cell lines was strongly anticorrelated with the level of DNA methylation at its promoter (Supplementary Fig. 5B), suggesting that the loss of DNA methylation itself can drive the expression. This notion is consistent with the published observation that treatment of HCT116 cells with 5-aza-deoxycytidine induces cGAS[64].

To ascertain the potential role of induced cGAS during senescence induced by loss of DNA methylation, we performed an shRNA knockdown (Fig. 5C). A cell-growth curve revealed no significant effect of *cGAS* knockdown on slow cell proliferation (Fig. 5D). However, the results of the SA-β-gal assay showed a noticeable decrease in staining in the conditions of *cGAS* knockdown compared to the control (Fig. 5E), and this finding was confirmed with an independent shRNA against *cGAS* (Supplementary Fig. 5C). RT-qPCR analysis confirmed the continuous suppression of *cGAS* after knockdown, and also revealed a

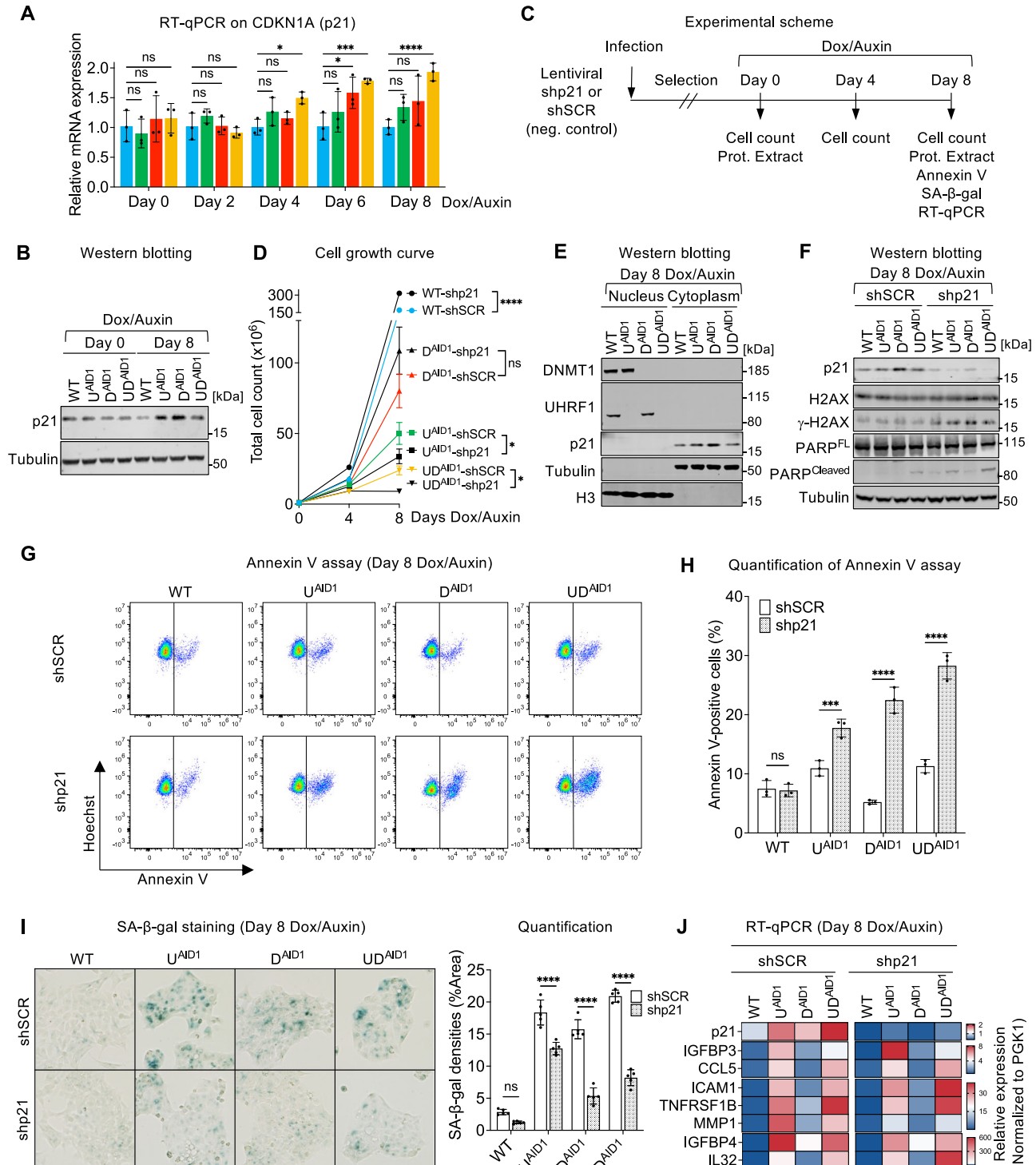

**Fig. 4 | p21 contributes to resistance against apoptosis during senescence caused by decreased DNA methylation. A** RT-qPCR analysis of *CDKN1A* (p21) mRNA levels across indicated days of Dox/Auxin treatment. $N = 3$ biological replicates. **B** Immunoblots for p21 in the indicated condition. **C** Experimental scheme of p21 depletion and assessment. **D** Total cell numbers at the indicated days of Dox/Auxin. $N = 3$ technical replicates. **E** Immunoblots for DNMT1, UHRF1, and p21 in the nuclear and cytoplasmic fractions of indicated lines at Day 8 of Dox/Auxin treatment. **F** Immunoblots of p21, H2AX, γ-H2AX, full-length (FL) PARP, and cleaved PARP in the indicated lines at Day 8 of Dox/Auxin treatment. **G** Representative results of Annexin V/Hoechst 33342 double staining in the indicated conditions.

**H** Quantification of Annexin V-positive cells at Day 8 of Dox/Auxin treatment. $N = 3$ technical replicates. **I** SA-β-gal staining (left panel) and quantification (right panel) of indicated lines. $N = 5$ fields of view. Scale bar is 50 μm. **J** RT-qPCR on *CDKN1A* (p21) and selected SASP genes in indicated conditions. Data of (**A**) are presented as mean ± SD and analyzed by one-way ANOVA with Dunnett's multiple comparisons test. Data of **D**, **H**, **I** are presented as mean ± SD and analyzed by two-way ANOVA with Sidak's multiple comparisons test. Heatmap data of **J** are presented as mean from three technical replicates. We use the following convention: *$p < 0.05$, ****$p < 0.001$, ****$p < 0.0001$, ns non-significant. Source data are provided as a Source Data file.

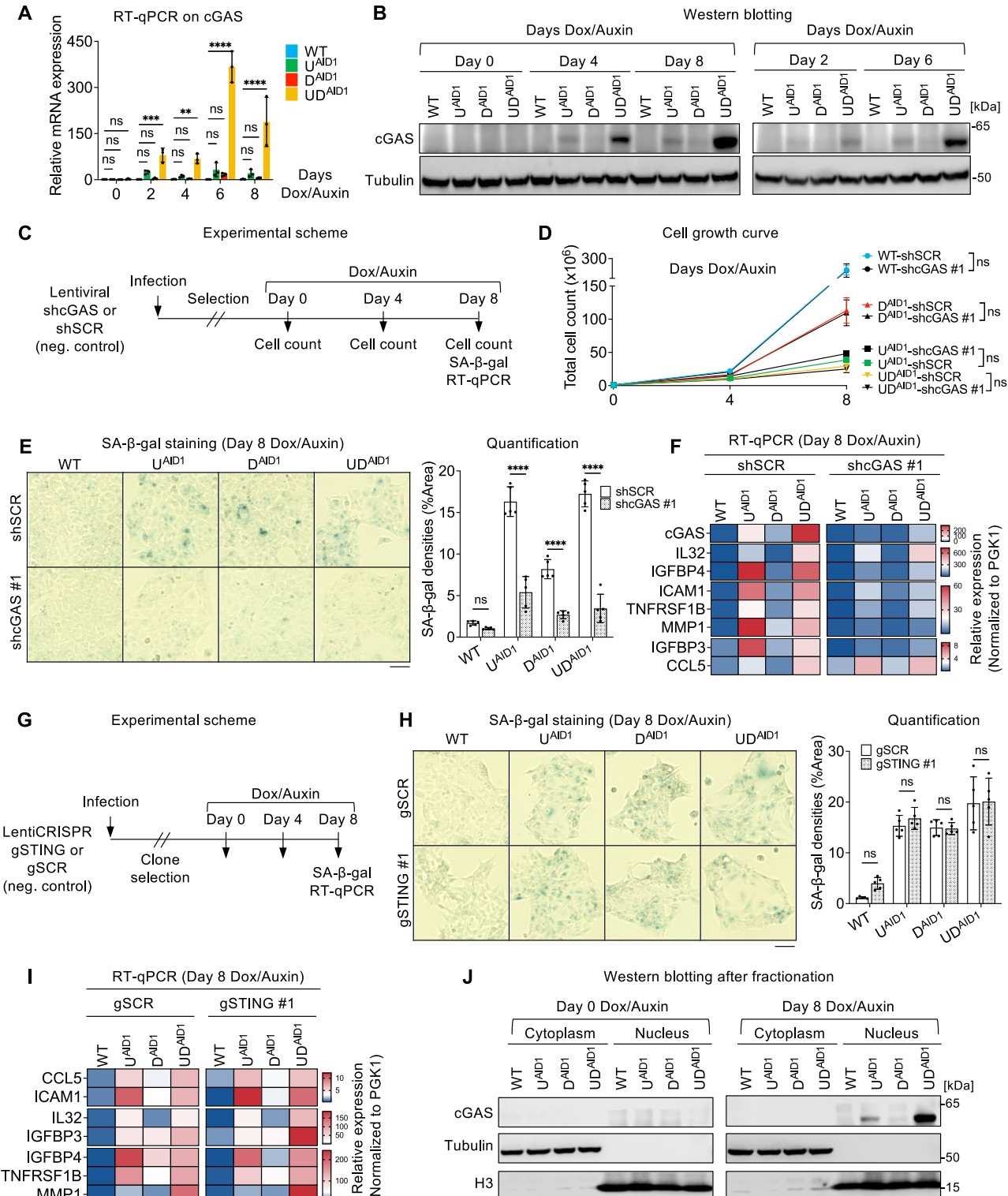

**Fig. 5 | cGAS is necessary for SA-β-gal positivity and SASP expression during senescence caused by loss of DNA methylation, and it acts independently of STING. A** RT-qPCR on *cGAS* in HCT116 lines upon Dox/Auxin treatment. N = 3 biological replicates. **B** Western blotting of cGAS protein in HCT116 lines at the indicated days upon Dox/Auxin treatment. **C** Scheme of cGAS knockdown experiments. **D** Total cell numbers at the indicated days upon Dox/Auxin treatment. *N* = 3 technical replicates. **E** SA-β-gal staining (left panel) and quantification (right panel) of the indicated lines. *N* = 5 fields of view. **F** RT-qPCR results of *cGAS* and selected SASP genes. **G** Scheme of *STING* knockout experiments. **H** SA-β-gal staining (left

panel) and quantification (right panel). *N* = 5 fields of view. **I** RT-qPCR results of selected SASP genes. **J** Western blotting of cGAS in cytoplasmic and nuclear fractions of HCT116 lines at the indicated days upon Dox/Auxin treatment. All scale bars are 50 μm. Data of **A**, **D**, **E**, **H** are presented as mean ± SD. Data of **A** are analyzed by one-way ANOVA with Dunnett's multiple comparisons test. Data of **D**, **E**, **H** are analyzed by two-way ANOVA with Sidak's multiple comparisons test. Heatmap data of **F**, **I** are presented as mean from three technical replicates. In all figures, we use the following convention: **p < 0.01, ***p < 0.001, ****p < 0.0001, ns: non-significant. Source data are provided as a Source Data file.

downregulation of the SASP genes that are typically induced after DNMT1/UHRF1 depletion, suggesting that cGAS activation contributes to the SASP gene induction (Fig. 5F). We also performed gain-of-function experiments by overexpressing cGAS or nuclear cGAS (NLS-cGAS) in untreated HCT116 cells (Supplementary Fig. 5D). This did not render cells positive for SA-β-gal (Supplementary Fig. 5E), showing that induction of cGAS is necessary but not sufficient for the phenotypes we observe.

Our next set of experiments addressed the possible role of STING in senescence induced by loss of DNA methylation. Our starting point was to examine the level of STING protein expression in the different cell types we used in our study. Western blotting showed that STING is indeed expressed in our HCT116 cells (Supplementary Fig. 5F), in agreement with previous findings[64]. To explore its function, we employed CRISPR/Cas9 to generate STING knockout (KO) clones in each of the degron lines, and then assessed the senescence phenotype in these KOs (Fig. 5G). The KO was efficient: no STING protein was detectable in any of the KO clones (Supplementary Fig. 5G). However, this genetic inactivation of STING had no effect on the induction of SA-β-gal positive cells (Fig. 5H and Supplementary Fig. 5H), nor on the expression of SASP genes (Fig. 5I).

A STING-independent contribution of cGAS to senescence has been reported, but the mechanisms remain elusive[65]. To test if cGAS is nuclear in our system, we performed a cellular fractionation on the degron cells after exposure to Dox/Auxin. We found that cGAS protein was accumulated in the nuclear fraction (Fig. 5J). These data agree with a very recent report also describing that cGAS is nuclear when expressed in HCT116 cells[66].

Collectively, our results reveal that chronic DNA methylation loss after UHRF1/DNMT1 depletion leads to the induction of cGAS, which itself contributes to the activation of SASP genes and increased lysosomal activity. This role of cGAS occurs in the nucleus and is independent of STING.

## Prolonged DNA demethylation also triggers cancer cell senescence in vivo

Having established that the depletion of UHRF1 and/or DNMT1 triggers senescence in cancer cells in vitro, we next sought to test if this also occurred in an in vivo setting. To perform these experiments, we turned to the AID2 degron system, in which the OsTIR1 (F74G) derivative can be activated by micromolar concentrations of the auxin analog 5-Ph-IAA, with minimal side effects in vivo[36]. We generated degron alleles of UHRF1 and/or DNMT1 in the HCT116 AID2 system, yielding the lines $U^{AID2}$, $D^{AID2}$ and $UD^{AID2}$ (Fig. 6A). We validated that 5-Ph-IAA treatment led to degradation of the tagged protein in these lines (Supplementary Fig. 6A), and consequently to a decrease of global DNA methylation (Supplementary Fig. 6B).

Cells from each condition (HCT116 $WT^{AID2}$, $U^{AID2}$, $D^{AID2}$, and $UD^{AID2}$) were transplanted into the flanks of nude mice, and formed observable tumors after 7 days. We then administered either saline or 3 mg/kg of 5-Ph-IAA via intraperitoneal (i.p.) injections daily for an additional 8 days. At the end of the experiment, the mice were sacrificed, and tumors were resected for a series of biological assessments (Fig. 6B).

We first verified the in vivo efficiency of the AID2 system. In all tumors we examined, treatment with 5-Ph-IAA led to a complete degradation of the cognate target (Fig. 6C), and this result was further confirmed by performing immunohistochemistry (IHC) for UHRF1 on tumor sections (Supplementary Fig. 6C, D). From these experiments, we conclude that the AID2 system permits efficient depletion of UHRF1 and/or DNMT1 in our xenograft experiments.

We next ascertained the consequences of UHRF1 and/or DNMT1 depletion in the xenografts. The first parameter we measured was tumor mass. All four cell lines generated tumors of similar mass when mice were injected saline (Supplementary Fig. 6E); in contrast the treatment with 5-Ph-IAA, leading to protein degradation, caused the

$UD^{AID2}$ xenografts to be significantly smaller than control xenografts, with a similar trend also observed in $D^{AID2}$ and $U^{AID2}$, albeit failing to reach statistical significance (Fig. 6D).

Our in vitro results led to the expectation that $U^{AID2}$, $D^{AID2}$, and $UD^{AID2}$ cells, treated with 5-Ph-IAA in vivo, may proliferate less than control cells and may enter senescence. We tested both predictions experimentally on tumor sections: the proliferation activity was estimated by Ki67 (a marker of proliferation) IHC, and the senescence by SA-β-gal staining.

Consistent with our in vitro findings, upon treatment, UHRF1 and/or DNMT1 degron xenografts showed a decreased signal of Ki67 compared to WT xenografts (Fig. 6E, F), indicating a cell proliferation defect. We next performed a whole-mount SA-β-gal assay on the tumors. After fixing and staining, tumor sections were counterstained with Nuclear Fast Red. Strikingly, the results revealed a dramatic increase of SA-β-gal positivity in $U^{AID2}$ and $UD^{AID2}$ xenografts, with a moderate increase of SA-β-gal positivity in $D^{AID2}$ xenografts upon treatment (Fig. 6G, H). In contrast, no SA-β-gal staining was detectable in xenografts from saline-treated mice (Supplementary Fig. 6F). Therefore, we conclude that loss of DNA methylation caused by degradation of UHRF1 and/or DNMT1 causes cancer cells to senesce in vivo.

The nude mice lack B and T lymphocytes, but they still possess components of the innate immune response, which can respond to a SASP[67]. To investigate this in our in vivo model, we used IHC staining for F4/80, a marker of macrophages. The results showed an increase of F4/80 positivity in tumors depleted of UHRF1 or UHRF1 and DNMT1, supporting the notion that the resulting senescent cells promote inflammation (Fig. 6I, J). Macrophages have a high lysosomal activity, raising the concern that they might generate a confounding SA-β-gal signal, however, a double SA-β-gal/F4/80 staining showed that the two signals were spatially distinct (Supplementary Fig. 6G). In addition, we detected positive p21 staining in the auxin-treated tumors, providing independent proof of the presence of senescent cells (Supplementary Fig. 6H).

To summarize, in comparison to control tumors, xenografts lacking UHRF1 and/or DNMT1 have lower proliferation, increased senescence, and macrophage infiltration, all of which may contribute to their reduced growth.

## Discussion

### Loss of DNA methylation induces senescence in multiple cancer cells

Through this study, we explored the consequences of lowering DNA methylation in cancer cells without promoting DNA damage. For this, we exploited degron cell lines for UHRF1 and DNMT1, two enzymes required for DNA methylation maintenance. We found that the loss of DNA methylation without DNA damage induced senescence in cancer cells.

HCT116 cells deprived of UHRF1 and/or DNMT1 present 4 hallmarks of cellular senescence: withdrawal from the cell cycle, enlarged nuclei, expression of SASP, and increased SA-β-gal activity (Figs. 1 and 2). Several arguments suggest that this phenotype is due to decreased DNA methylation, as opposed to other effects of removing UHRF1 and/or DNMT1. First, our rescue experiments with 9 point-mutants of UHRF1 and DNMT1 show a complete overlap between the ability to sustain DNA methylation and the ability to prevent senescence (Supplementary Fig. 1). Second, removing UHRF1 causes a more rapid loss of DNA methylation than removing DNMT1[29], and it also causes a more rapid entry into senescence. Third, treatment with a last-generation inhibitor of DNMT1 that does not induce DNA damage[37] also causes cancer cells to become senescent (Fig. 1). In parallel to our work, a degron approach has been recently applied by other authors to DNMT1 in DLD-1 colorectal cancer cells[68]. This study shows that DNMT1 depletion leads to impaired proliferation with a G1-phase

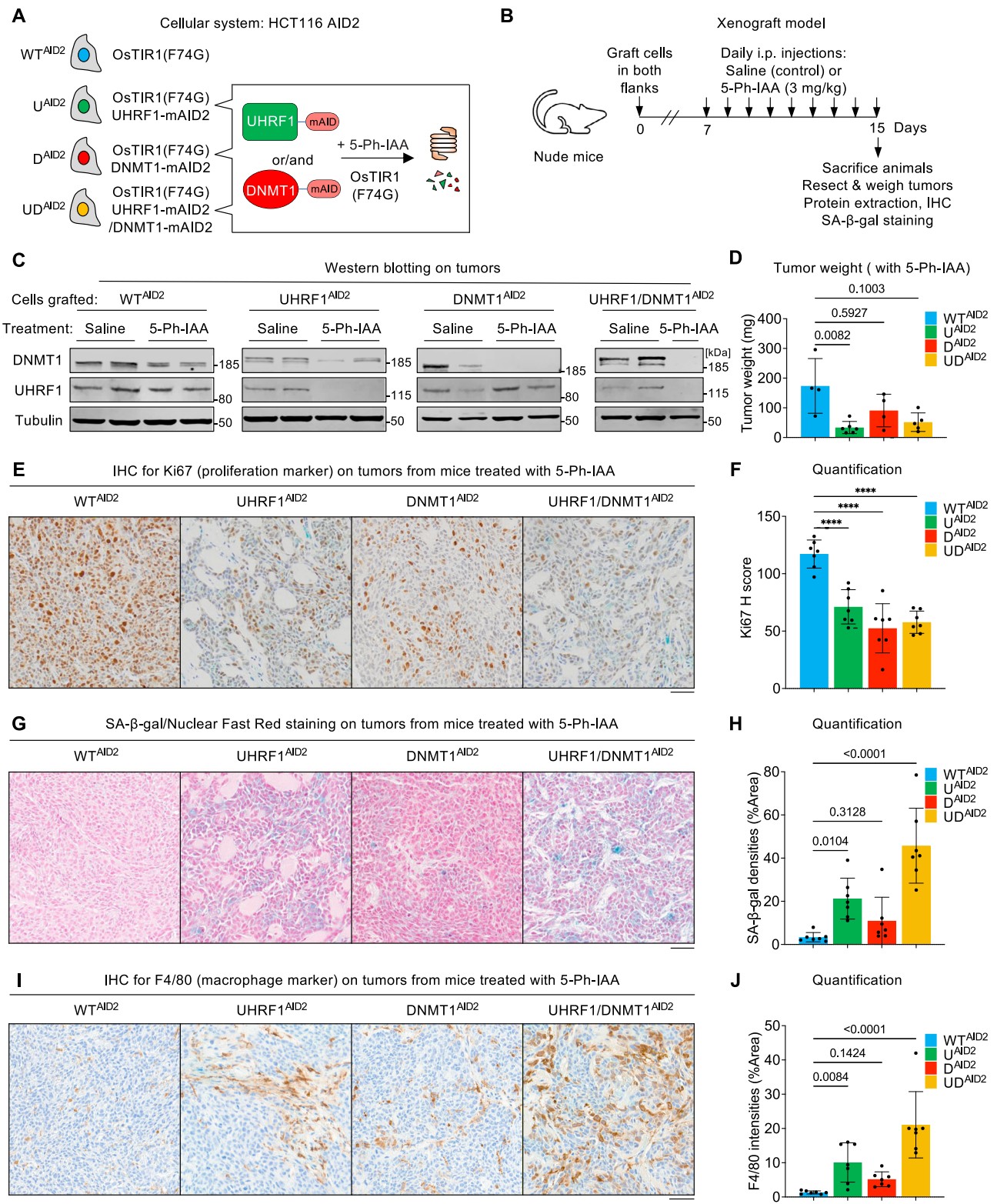

**Fig. 6 | Senescence caused by loss of DNA methylation also occurs in vivo. A** The auxin degron system in HCT116 cells constitutively expressing OsTIR1(F74G). **B** Scheme of the xenograft experiments. **C** Western blot validating the degradation of UHRF1 and/or DNMT1 upon 5-Ph-IAA treatment of mice. **D** Tumor weight in mice treated with 5-Ph-IAA. $N = 4$ tumors for WT$^{AID2}$, 6 tumors for U$^{AID2}$, 4 tumors for D$^{AID2}$, and 5 tumors for UD$^{AID2}$. **E** Immunohistochemistry (IHC) staining of Ki67 in indicated tumors. **F** Quantification of Ki67 IHC using H score. $N = 7$ fields of view at 400× original magnification. **G** Representative images of SA-β-gal staining in indicated tumors.

Nuclear Fast Red (NFR) was used for counterstaining. **H** Quantification of SA-β-gal staining. $N = 7$ fields of view at 300× original magnification. **I** IHC staining of F4/80 in indicated tumors. **J** Quantification of F4/80 IHC. $N = 7$ fields of view at 400× original magnification. All scale bars are 50 μm. All data are presented as mean ± SD. Data of **D, H, J** are analyzed by Kruskal–Wallis test and Dunn's multiple comparison test. Data of **F** are analyzed by one-way ANOVA and Dunnett's multiple comparisons test. We use the following convention: ****$p < 0.0001$. Source data are provided as a Source Data file.

increase, S-phase decrease, and few sub-G1 cells; all of these observations are compatible with an entry into replicative senescence.

The senescence phenotype in cells with decreased DNA methylation is not restricted to a specific cell line, or a specific tumor type, as we have observed it in multiple cancer lines originating from the colon, lung, breast, and cervix (Fig. 1). It will be instructive in future studies to determine if certain cancer lines deviate from this response and if so, why.

The senescence phenotype triggered by loss of DNA methylation is highly penetrant in HCT116 and other cancer cell lines, yet has not been reported in earlier publications. This likely results from prior studies using agents such as 5-aza-C or 5-aza-dC, which induce DNA damage, to research DNA demethylation. High doses of the drugs generate unsustainable DNA damage, resulting in apoptosis. Using smaller amounts of the drug may lead to a partial or delayed senescence phenotype, which could be challenging to detect if it is masked by non-senescent cells of if the observation period is not sufficiently long. Other studies have examined loss of DNA methylation after RNAi or shRNA approaches on UHRF1 or DNMT1[38,69,70], and have not reported senescence. In retrospect, our results suggest that these approaches have likely imposed a positive selection on the cells with the least depletion, possibly obscuring the senescent phenotype. Finally, HCT116 cells that are genetically hypomorphic for DNMT1[71,72] are not reported to be senescent, but this could be because they retain a sufficient level of DNMT1 activity and/or because they have adapted genetically and/or epigenetically. In summary, we believe the degron approach has unmasked a phenotype that was difficult to detect with earlier tools.

Compared to previous strategies, the degron approach also permits more precise kinetic studies, allowing us to order in time the different elements of the cellular response to DNA demethylation. For instance, we report that, while the activation of an interferon response signature is an early event (within 2 days), the full induction of interferon lambda is more delayed (Fig. 2). These observations pave the way for further studies in the future, to fill in additional mechanistic details underlying our findings.

The consequences of removing UHRF1 from normal cells differ from what we observe in cancer cells. For instance, genetically removing UHRF1 from liver hepatocytes stops their proliferation[73]. However, this phenotype is due to replication stress, DNA damage, and S-phase arrest[74,75], all of which are absent in the degron cancer cells. In mouse neural stem cells, the genetic inactivation of UHRF1 does not affect proliferation, survival, or differentiation. Instead, it induces the reactivation of intracisternal type A particle (IAP) repeats, eventually leading to apoptosis[76]. To extend our work in the coming years, a broad endeavor will be to explore what mechanisms underlie the different responses of normal and cancer cells to the loss of DNA methylation, and to determine if our findings could relate to the changes of DNA methylation that occur in normal cells during organismal aging[77].

## Non-canonical mechanisms underpin senescence induced by loss of DNA methylation

Senescence induced by loss of DNA methylation differs in several ways from other instances of senescence previously reported in normal and cancer cells.

First, many of the known senescence-inducing stresses, such as oncogenic activity, DNA damage, or metabolic stresses, converge on p53 and/or Rb[78], but these two factors are dispensable for the loss of DNA methylation to trigger senescence (Fig. 3).

Another salient result is that DNA damage, which plays a central role in many senescence types, is unlikely to be involved in senescence triggered by DNA demethylation (Fig. 3). Given that high levels of retrotransposition cause DNA damage[79], this result also implies that retrotransposition is either mild or absent in cells with decreased DNA

methylation. This may be explained in part by the observation that DNA damage itself can activate repeated elements[80]. Therefore, a 5-azacytidine treatment would cause a stronger reactivation of repeated elements than DNA demethylation alone.

An additional difference is that, in cells with lower DNA methylation, cytosolic p21 plays a protective role against apoptosis (Fig. 4), at variance with nuclear p21 inhibiting the cell cycle in canonical senescence models[78].

A last distinction concerns the role of cGAS. Cytosolic chromatin fragments can trigger senescence[61,62], a phenomenon implying cytosolic cGAS, which becomes catalytically active and produces cyclic GMP-AMP (cGAMP), thereby stimulating STING. In cells with decreased DNA methylation, cGAS is nuclear and STING is not required for senescence (Fig. 5), so different processes must be at play. In particular, the results argue that cytoplasmic DNA originating from transposable elements does not have a major contribution to the phenotypes we observe, unlike what is seen in replicative senescence[79]. A recent study found that chromatin-bound cGAS can recruit the SWI/SNF complex at specific chromatin regions to regulate gene expression[66], thus a similar process might be at play in our system. It has also been reported that UHRF1 inhibits the activity of nuclear cGAS[81], and this may also contribute to our observations.

Chromatin-based mechanisms underlying SASP expression and senescence are regularly being discovered. For instance, spurious intragenic transcription participates to senescence[82], while the histone chaperone HIRA plays a histone-independent role in SASP induction[83]. In follow-up studies, it will to determined if these phenomena contribute to the senescence phenotypes we observed upon DNA methylation decrease.

There are other interesting possibilities that could be pursued in the future to precise how loss of DNA methylation causes cellular senescence in cancer cells. We observe an increase of H3K9Ac, an "active" mark, in the degron cells (Supplementary Fig. 2), while increases in Pol II elongation speed have been shown to contribute to aging in model species[84], warranting further investigation into a possible contribution of increased transcription to senescence in our system. In addition, decreasing DNA methylation in HCT116 and DLD1 cells has been shown to alter their 3D genome organization[68,85,86], raising the question of whether is reorganization is a cause, consequence, or a parallel event to senescence caused by loss of DNA methylation. Lastly, decreased DNMT1 expression in hTERT RPE1 cells causes centromeric hypomethylation[87], which could conceivably also occur in our system, even though aneuploidy does not appear to be prevalent.

## Therapeutic implications

Our xenograft experiments (Fig. 6), show that cancer cells with decreased DNA methylation undergo senescence in vivo. Inducing senescence in tumor cells is emerging as a promising therapeutic avenue[78,88]. Not only does this strategy suppress their proliferation, but the accompanying SASP also has a paracrine pro-senescent effect on neighboring tumor cells that may have escaped treatment. Two added benefits are that the SASP can make the vasculature more permeable to drugs[89], and that it activates an inflammatory response that can promote clearing by the immune system[88]. Nevertheless, inducing senescence in tumors may be a double-edged sword, as senescent cells can become immunosuppressive, induce an Epithelium to Mesenchyme Transition (EMT) in non-senescent cells, or promote angiogenesis[88]. It may be possible to circumvent these problems by combining the induction of senescence by loss of DNA methylation with a senolytic treatment, to effectively clear the senescent cancer cells[88].

Several existing cancer therapies can be used to promote cancer cell senescence, including the use of genotoxics[90]. Our data show that an additional option to achieve this goal is to decrease DNA methylation, without causing DNA damage. A clinically important implication

of our results is that this strategy should be applicable to the many tumors lacking p53 and/or Rb.

Molecularly, decreasing DNA methylation without DNA damage can be achieved with DNMT1 inhibitors that do not incorporate into DNA, such as the recently described GSK3685032[37], or its derivatives[91]. However, our degron experiments show that UHRF1 inhibition has a faster and/or stronger effect than DNMT1 inhibition, which likely stems from DNMT1-independent roles of UHRF1[29]. This suggests that the development of UHRF1 inhibitors may be a promising strategy for inducing senescence without DNA damage in cancer cells. Our work identifies domains of UHRF1 that are essential for preventing senescence of cancer cells (Supplementary Fig. 1). These domains are potentially druggable, and UHRF1 has already been used for chemical screens in vitro[92–95], but an efficient and specific inhibitor has yet to emerge. In some instances, targeted protein degradation by a proteolysis targeting chimera (PROTAC) is superior to chemical inhibition[96], and this approach could be explored for UHRF1 as well.

As for any other drug, a prerequisite for UHRF1 inhibitors to be useful is the possibility to treat cancer cells without harming healthy cells. This might be solved by addressing the drugs specifically to the tumor. Alternatively, given that many tumors highly overexpress UHRF1[38,97], they might possibly have developed a higher reliance on UHRF1 than normal cells, which would open a possible therapeutic window. These and other questions can be addressed in the future on the basis of the findings reported here.

## Methods

### Ethics statement
This study complies with all relevant ethical regulations. Mice were maintained in laminar-flow boxes under standard conditions (standard diet and water ad libitum) in Plateforme du Petit Animal du CRCL (P-PAC) specific pathogen-free animal house (Lyon, France). Experiments were performed according to animal care guidelines of European and French laws. Protocols were approved by the French Ministry of Research and Innovation (ethical permit APAFIS #49702-202404181412449), in accordance with European Union guidelines and with the local animal ethics committee (CEEA-010 – ACCeS).

### Cell lines and cell culture
The HCT116 cells are male. Their derivative expressing OsTIR1 under a doxycycline-inducible promoter was developed and published by the Kanemaki lab[33], and we described in a previous publication how we used them to generate UHRF1, DNMT1, and UHRF1/DNMT1 degron cells, and how we rescued these cells genetically with WT or mutant constructs[29]. We also used DLD1 cells (male) constitutively expressing the OsTIR1(F74G) improved variant[36] to generate degron alleles of UHRF1, DNMT1, or both, in this AID2 system[29].

For the current publication, we generated a new series of degron lines in the HCT116 AID2 background. The cells were transfected with CRISPR-Cas9 and donor plasmids using Lipofectamine 2000. After 700 μg/mL G418 or 100 μg/mL hygromycin selection, we obtained HCT116 AID2 lines (UHRF1-mAID2, DNMT1-mAID2, and UHRF1-mAID2/DNMT1-mAID2). Other human cancer cell lines, including HT29 (female), A549 (male), MCF7 (female), HeLa (female), MDA-MB-468 (female), SaoS2 (female), were acquired from the ATCC. The HCT116 p53 null (p53$^{-/-}$) cell line was a gift from Bert Vogelstein[98].

To achieve degradation of the AID-tagged protein, AID1 cells were seeded and incubated with 2 μg/mL doxycycline (Dox) and 20 μM auxinole for 24 h. Afterwards, the medium was replaced with a fresh medium containing 2 μg/mL Dox and 500 μM indole-3-acetic acid (IAA, auxin). For AID2 cells, they were seeded one day prior to the addition of medium containing 1 μM 5-Ph-IAA, an IAA derivative. The medium with either Dox/Auxin or 5-Ph-IAA was replaced every two days.

HCT116, HT29, and SaoS2 cells were cultured in McCoy's 5 A medium supplemented with 10% FBS, 2 mM L-glutamine, 100 U/mL penicillin, and 100 μg/mL streptomycin. DLD1 cells were cultured in RPMI-1640 medium with the same supplements as above. A549, MCF7, HeLa, and MDA-MB-468 cells were cultured in DMEM medium, also supplemented identically. All cultures were incubated at 37 °C under a humidified atmosphere containing 5% $CO_2$. Cells were tested monthly for Mycoplasma infection. The cell identity was confirmed by short tandem repeat (STR) analysis.

As a positive control for apoptosis induction, HCT116 WT cells were treated with 40 μg/mL 5-Fluoro-Uracil (5-FU) for 2 days. As a positive control of acute DNA damage induction, HCT116 WT cells were treated with 2 mM hydroxyurea (HU) or 10 μM etoposide (Eto) for 4 h. As a positive control of persistent DNA damage induction, HCT116 WT cells were treated with 10 μM etoposide for 24 h, followed by replacement with fresh medium for an additional 5 days.

### Plasmid construction and viral infection
shRNA targeting Scramble (negative control), cGAS, p21, or ORF1 was cloned into the pLKO.1-blast vector (Addgene #26655). Plasmids were generated using PCR, restriction enzymes, or Gibson Assembly (NEB) Cloning techniques. The oligonucleotide sequences are inserted into the pLKO.1-blast vectors are available in Supplementary Data 2. The vector pLenti X2 Blast/shp16 (w112-1) (Plasmid #22261) and pLenti X2 Blast/pTER shLUC (w618-1) (Plasmid #20962) were from Addgene.

Human papillomavirus type 16 (HPV16) E6-E7 combined genes were amplified from the template vectors (pLXSN E6-E7, Addgene #52394) and then cloned into pLenti6.2/V5-DEST lentiviral vector (Invitrogen). gRNA targeting STING and control were gifts from Nicolas Manel (Institute Curie, FR). The complete list of gRNA sequences is shown in Supplementary Data 2. The wildtype cGAS (Addgene #102612) and NLS-cGAS (Addgene #127656) overexpression plasmids were also obtained from Nicolas Manel. All plasmids underwent sequencing prior to their utilization.

For viral infections, HCT116 WT and degron lines were plated overnight, then infected with retroviruses in the presence of 6 μg/mL polybrene (Sigma-Aldrich) for 16 h. After 48 h, infected cells were selected with 10 μg/mL blasticidin for 7 days or 1 μg/mL puromycin for 2 days.

### Cell counting
Cells were initially seeded at 0.125 million per well in 6-well plates and cultured in medium supplemented with 2 μg/mL Dox and 20 μM auxinole. On the following day (Day 0), the medium was refreshed with one containing 2 μg/mL Dox and 500 μM auxin. Cell counts were performed on Day 4, after which cells were passaged at a density of 0.25 million, and again on Day 8. The Dox/Auxin-containing medium was renewed every two days. A BioRad TC20 automated cell counter was employed to determine total cell numbers.

### Protein extraction
Cells were harvested after trypsinization, washed twice with PBS, and lysed with RIPA buffer (Sigma-Aldrich) with protease inhibitor cocktail (Roche) and phosphatase inhibitor cocktail 3 (Sigma-Aldrich) for 20 min on ice, and then sonicated with series of 30 s ON / 30 s OFF for 5 min on a Bioruptor device (Diagenode) and centrifuged at $16,000 \times g$ for 15 min at 4 °C. The supernatant was collected and quantified by Pierce BCA protein assay kit (Thermo Fisher Scientific).

### Cell fractionation
Cell pellets (5 million) were gently lysed in 500 μL of hypotonic buffer (20 mM of Tris-HCl (pH 7.6), 3 mM of $MgCl_2$, 10 mM of NaCl) supplemented with protease inhibitor cocktail (Roche) and phosphatase inhibitor cocktail 3 (Sigma-Aldrich). Incubate on ice for 20 min. Add 25 μL detergent (10% NP-40) and vortex for 10 s at highest setting. Following centrifugation for 10 min at $1000 \times g$ at 4 °C, the supernatants were collected as the cytoplasmic fraction. Pellets were

washed twice with hypotonic buffer and then lysed by 200 μL RIPA buffer. After incubation on ice for 30 min, the lysates were centrifuged for 30 min at 16,000 × g at 4 °C. The supernatants were then collected as the nuclear fraction.

## Western blot

Equivalent amounts of protein extract per sample were mixed with NuPage 4X LDS Sample Buffer and 10X Sample Reducing Agent (Thermo Fisher Scientific) and denatured at 95 °C for 5 min. Samples were resolved on a pre-cast SDS-PAGE 4-12% gradient gel (Invitrogen) and transferred to a nitrocellulose membrane (Millipore). The membranes were blocked in 5% (w/v) non-fat milk in PBST buffer (PBS with 0.1% Tween 20) or 5% (w/v) bovine serum albumin (BSA) in TBST buffer (TBS with 0.1% Tween 20), then incubated overnight at 4 °C with the appropriate primary antibodies diluted in 5% milk PBST or 5% BSA TBST. After washing with PBST or TBST, the membranes were treated with fluorophore-linked or HRP-conjugated secondary antibodies and visualized using the LI-COR Odyssey-Fc imaging system (LI-COR).

## BrdU labeling and flow cytometry

To analyze the cell cycle, HCT116 cells were seeded in a 6-well plate and treated with Dox/Auxin for up to 8 days. The cells collected at Day 0, Day 4, and Day 8 were treated with 10 μM 5-bromo-2′-deoxyuridine (BrdU) (Sigma-Aldrich) for 1 h at 37 °C in a CO$_2$ incubator. Cells were washed, trypsinized, and resuspended in 750 μL of PBS before adding 2250 μL of ice-cold ethanol for fixation (yielding a final volume of 3 mL with 75% ethanol). The fixed cells were incubated for at least 30 min at −20 °C prior to flow cytometry analysis. Samples were denatured in 2 N HCl for 15 min at room temperature, followed by two washes in PBS + 1% BSA. Samples were then incubated in 200 μL of anti-BrdU-FITC antibody (BD Biosciences) in PBS + 1% BSA for 60 min. After a washing step, samples were then incubated in PBS containing propidium iodide (PI) (1:500, Invitrogen) with 150 μg/mL RNAse A (Qiagen) overnight at 4 °C in the dark. Cell cycle profiles were measured by flow cytometry using the Cytoflex SRT (Beckman Coulter). Data were analyzed with the FlowJo v10.4 software. Cell populations were gated by size using SSC-A and FSC-A parameters. For single-cell resolution, we employed FSC-H versus FSC-A gating. All gated information was supplied for cell population analysis.

## EdU incorporation assay

Cell proliferation was assessed using the Click-iT EdU Alexa Fluor 647 imaging kit (Invitrogen). Cells were seeded in 4-well plates two days before fixation and incubated with 10 μM 5-ethynyl-2′-deoxyuridine (EdU) for 18 or 20 h at 37 °C. After fixation with 3.7% paraformaldehyde for 15 min at room temperature and permeabilization with 0.5% Triton X-100 for 20 min at room temperature, the Click-iT reaction was performed according to the manufacturer's protocol. Nuclei were counterstained with DAPI, and images were acquired using Leica DMI6000 (Leica Microsystems. EdU-positive cells were quantified using ImageJ software (v2.14.0) by calculating the ratio of EdU-positive nuclei to total nuclei in at least 4 random fields per condition.

## Colony formation

HCT116 WT and degron cells were pretreated with Dox/Auxin for 8 days prior to seeding. Cells were then plated in 6-well plates at a density of 2500 cells/well and incubated for 6 days at 37 °C in a humidified atmosphere with 5% CO$_2$ without auxin treatment. After incubation, colonies were washed twice with PBS and fixed with absolute methanol for 15 min at room temperature. Fixed colonies were stained with 1% crystal violet solution (w/v in water; Sigma) for 20 min at room temperature. Excess stain was removed by washing with PBS, and plates were air-dried. Colonies were counted using ImageJ. Colony formation efficiency was calculated as the percentage of colonies formed relative to the number of cells initially seeded.

## TUNEL assay

HCT116 cells were seeded on coverslips two days before fixation. The cells were fixed with 4% paraformaldehyde for 15 min, then permeabilized using 0.25% Triton X-100 in PBS for 20 min at room temperature. For the TUNEL assay, the following steps were performed using the Click-it Plus TUNEL Assay according to the manufacturer's protocol (Invitrogen). For a positive control, cells were treated with DNase I (Invitrogen) for 30 min at room temperature. Cells were additionally stained for DNA using 1× Hoechst 33342 (Life Technologies). Cells were imaged using Leica DMI6000 (Leica Microsystems).

## Annexin V assay

Cells adhered to the dishes, and those detached in the supernatant were collected for apoptosis assays. Cell pellets were then resuspended in Annexin Binding buffer (10 mM HEPES, 140 mM NaCl, 2.5 mM CaCl$_2$, pH 7.4). The following steps were performed using the Annexin V Conjugates for Apoptosis Detection kit (Invitrogen). Cells were stained with 0.1% (v/v) Annexin V and 1× Hoechst 33342 (Life Technologies) solution. Stained cell suspensions were incubated in the dark for 15 min at room temperature. 400 μL of 1× binding buffer was added to each tube, and samples were then immediately analyzed by flow cytometry using the Beckman Coulter Cytoflex SRT.

## Immunofluorescence

Cells were fixed in 4% paraformaldehyde at room temperature for 15 min. After fixation, cells were permeabilized with 0.5% Triton X-100 in PBS for 10 min at 4 °C, then washed with PBS. Cells were blocked with 1% BSA in PBS at room temperature for 30 min, then incubated with γ-H2AX antibody (Sigma-Aldrich 05-636, 1:1000) or cGAS antibody (CST 79978, 1: 1000) for 1 h at room temperature. After washing three times with PBST, cells were incubated with secondary antibodies conjugated with Cy5 for 1 h at room temperature, washed with PBST three times, and finally mounted with ProLong Diamond Antifade Mountant with DAPI (P36961, Thermo Fisher Scientific). Images were obtained using Leica DMI6000 (Leica Microsystems) and MetaMorph Leica v6.1 software.

## RNA extraction and quantitative reverse transcription PCR (RT-qPCR)

Total RNA was extracted from cells with the RNeasy Plus Mini kit (Qiagen) according to the manufacturer's instructions and quantified using Nanodrop or Qubit RNA BR Assay kit on the Qubit 2.0 Fluorometer (Thermo Fisher Scientific). One micogram of total RNA was reverse transcribed using SuperScript IV Reverse Transcriptase (Thermo Fisher Scientific) and random primers (Promega). RT-qPCR was performed using Power SYBR Green (Life Technologies) on Applied Biosystems QuantStudio 6 Pro. Housekeeping gene PGK1 was used for normalization. All the primer sequences for RT-qPCR were listed in Supplementary Data 2.

## Senescence-associated β-galactosidase (SA-β-gal) staining

SA-β-gal staining was performed with the Senescence Cells Histochemical Staining Kit (Sigma–Aldrich, cat # CS0030-1KT). Briefly, monolayered cells in 6-well plates were incubated with fixation buffer containing 2% formaldehyde for 7 min at room temperature, followed by 37 °C overnight incubation in staining solution, supplemented with 1 mg/mL of 5-bromo-4-chloro-3-indolyl-β-D-galactopyranoside (X-gal). Plates with staining solution were kept overnight at 37 °C without CO$_2$ until the cells were stained blue. The plates were sealed with parafilm to prevent them from drying out. The stained cell images were taken using LEICA DMi1 (Leica Microsystems).

## Conditioned medium (CM) collection, CM culture, and enzyme-linked immunosorbent assay (ELISA)

Conditioned media from four conditions were prepared. Conditioned medium from HCT116 AID1 cells: WT and degron cells were cultured in

McCoy's 5 A medium supplemented with 10% FBS and Dox/auxin for 8 days. Cells were then washed twice with PBS, followed by incubation in FBS-free McCoy's 5 A medium for 20 h. Conditioned medium from the interferon response positive control: HCT116 WT cells transfected with PolyI:C (1 µg/mL, Invivogen) using Lipofectamine RNAiMAX (Invitrogen) for one day underwent a wash step and medium replacement before incubating in FBS-free McCoy's 5A medium for another 20 h. Conditioned medium from interferon response negative control: WT cells were cultured in McCoy's 5 A medium without FBS for 20 h. CM from DNA damage induced senescence: WT cells were treated with 1 µM etoposide for 24 h, followed by a 3-day fresh medium culture before switching to FBS-free McCoy's 5 A medium for an additional 20 h.

Serum-free conditioned media were collected after counting cell numbers to ensure equal volumes from equivalent cell counts. Media was centrifuged at $300 \times g$ for 5 min to remove cellular debris and filtered through a sterile 0.2 µm filter (Thermo Fisher Scientific).

For CM culture, WT cells were culture in CM supplemented with an additional FBS for 24 h. For ELISA, CM from WT, PolyI:C transfected WT, Dox/Auxin treated WT, and degron cells were 10× concentrated using Amicon Ultra centrifugal filter (Millipore). The ELISA assay was performed according to the manufacturer's protocol (Thermofisher, cat #88-7296) to detect the presence of human IL-29 (Interferon lambda 1). Detection was performed using AMR-100 Microplate Reader at 450 nm (Hangzhou Allsheng).

## Genomic DNA extraction and Luminometric methylation (LUMA) assay

Genomic DNA was extracted by Monarch Spin gDNA extraction kit following the manufacturer's protocol. LUMA analysis was conducted following standard procedures. In brief, to assess global CpG methylation, 500 ng of genomic DNA was digested with Msp I + EcoR I and Hpa II + EcoR I (New England Biolabs) in parallel reactions. EcoR I was included as an internal reference. CpG methylation percentage is defined as the Hpa II/Msp I ratio. Samples were analyzed using Pyro-Mark Q24 Advanced pyrosequencer (Qiagen).

## Xenograft Mouse models

Six-week-old female Rj:NMRI-Foxn1 nu/nu mice were acquired from Janvier Labs and maintained under standard conditions (standard diet and water ad libitum) at 23 °C with 12 h light and 12 h dark cycles, in a specific pathogen-free (SPF) animal facility; they were randomly selected and acclimated a week before performing experiments. As sex differences were not the focus of this study, we used only female mice to avoid any aggressive behavior of males, allowing to obtain more stable group housing, less stress, and reducing variability in line with the 3Rs principle. To form xenografts, a suspension of HCT116 cells in 50% Matrigel (Corning #354234) was injected subcutaneously into the right and left flanks of each mouse. Large (L) and small (l) length of each tumor were measured using a caliper, and the tumor volume was calculated using the following formula: Volume = ½ (L × l²). Before drug treatment, we first let the tumors engraft until they reached a volume of ~60 mm³. The mice were injected intraperitoneally daily for 8 days with 5-Ph-IAA (3 mg/kg) or with NaCl 0.9%. Animal work was conducted according to the ethical guidelines applicable in France, in a duly authorized animal facility (P-PAC, agreement number D693880202). Our project received the agreement number APAFIS #49702-202404181412449 v3 from the French Ministry of Education and Research. In accordance with ethical guidelines, the maximal permitted tumor burden was set at 1500 mm³. No animals exceeded this limit during the study.

## Tumor protein extraction

Tumors collected from the xenograft mouse models were cut and placed in a RIPA buffer (Sigma-Aldrich) with protease inhibitor cocktail (Roche) and 0.05% PMSF. Samples were further homogenized using Precelly's ceramic beads 1.4 mm (13113-325) and following the manufacturer's protocol for protein extraction with the Precelley's 24 Dual Lysis and Homogenization (P002511-PEVT0-A.0, Bertin Technologies).

## Whole-mount SA-β-gal staining

At resection, tumors were snap frozen and kept at −80 °C until the assay. After thawing, tumors were fixed and SA-β-gal whole mount staining was performed overnight according to the procedures described in the Senescence Beta-Galactosidase Cell Staining Kit (Cell Signaling Technology #9860). All incubations and washes were performed with fresh and filtered solutions. Stained tissues were then washed twice in PBS and post-fixed with 4% paraformaldehyde before paraffin embedding for sectioning. 8 µm sections were cut and counterstained with Nuclear Fast Red (Sigma-Aldrich #N30-20-100mL).

## Immunohistochemistry (IHC) staining and scoring

For IHC staining, 4 µm fixed tumor sections were incubated for 15 min at 95 °C in a solution of deparaffinization and antigen retrieval at pH 6 or pH 9. Then they were rinsed and incubated 20 min at room temperature with primary antibodies against UHRF1, Ki67, F4/80 or p21. The revelation in brightfield was performed by "Bond Polymer Detection" (Leica DS9800). Sections were lightly counterstained with hematoxylin, then dehydrated and cleared in graded alcohol and Ottix plus (MM-France), and finally covered with glass slips.

An IHC score was calculated by summing the products of staining intensities(0-3) and percentage of stained cells in each intensity(0-100); H scores ranged from 0 to 300. The intensity level was evaluated by the IHC profiler plugin in ImageJ software (v2.14.0). 0 points were assigned for negative, 1 for low positive, 2 for positive, and 3 for high positive. Each section was assayed for seven independent high magnification (×400) fields to get the average scores.

## Bulk RNA sequencing, data processing, transposable element (TE) analysis, and Virtual Inference of Protein-activity by Enriched Regulon analysis (VIPER)[57]

Sequencing libraries were prepared from poly-A-selected RNA following the manufacturer's protocol (Illumina, San Diego, CA, USA). Paired-end sequencing (150 bp) was performed on an Illumina NovaSeq 6000 platform (Illumina, San Diego, CA, USA). Raw RNA-seq reads were assessed for quality using FastQC (v0.11.9) and subsequently trimmed to remove adapter sequences and low-quality bases using Trim Galore (v3.0) with default parameters. Processed reads were aligned to the human reference genome (GRCh38/hg38) using STAR (v2.6.1d) with standard alignment parameters. Gene-level read counts were generated using featureCounts (v1.5.0-p3) from the Subread package, with annotations derived from the reference genome. Normalization and differential expression analysis were performed using DESeq2 (v1.38.3) in R. Genes with an adjusted p-value threshold of <0.05 were considered statistically significant. Expression levels were further normalized and reported as transcripts per million (TPM) for downstream analyses.

Gene set enrichment analysis was performed using GSEA (v4.1.0) and default parameters (1000 permutations, permutation type = gene_set). Selected gene sets derived from the Molecular Signatures Database (MSigDB) Hallmark and Kyoto Encyclopedia of Genes and Genomes (KEGG) were used to determine the enrichment of senescence-related genes. GSEA computes a normalized enrichment score (NES) that reflects the degree to which a test gene set is over-represented in upregulated or downregulated genes of another gene set. GSEA results with a false discovery rate (FDR) lower than 0.05 and an absolute NES larger than 1 were considered statistically significant.

Transposable element (TE) expression was quantified using TEcount (v2.2.3) from the TEtranscripts package[41]. Quality-filtered bulk RNA-seq reads (trimmed with Trim Galore v3.0) were aligned to the GRCh38/hg38 reference genome, which was augmented with TE

annotations from the RepeatMasker track (UCSC Genome Browser), using STAR (v2.6.1d). TEcount was then employed to generate raw TE counts, using the non-stranded mode to account for library preparation protocols. Multi-mapping reads were distributed probabilistically to account for ambiguous alignments across repetitive regions. Differential TE expression analysis was performed on raw count data using DESeq2 (v1.38.3). Counts were normalized using the DESeq2 median-of-ratios method, and statistical testing was conducted with an adjusted $p$ value threshold of <0.05 to define significant differential expression.

Protein activity was inferred using the msVIPER algorithm from the viper R package. VIPER estimates protein activity by assessing the enrichment of transcription factor (TF) regulons—sets of target genes regulated by a given TF—in gene expression profiles. For this analysis, we used a colon adenocarcinoma (COAD)-specific regulatory network derived from The Cancer Genome Atlas (TCGA), obtained via the aracne. Networks R package. This network was constructed using ARACNe (Algorithm for the Reconstruction of Accurate Cellular Networks), which infers TF-target interactions by computing mutual information between genes and applying a data-processing inequality step to eliminate indirect regulatory relationships. The msVIPER function was applied to the bulk RNA-seq to compute protein activity scores. The algorithm evaluates the statistical overrepresentation of regulon target genes in the input expression profile, normalizes scores to account for differences in regulon size and expression variance, and generates an activity score for each protein. To determine significance, we performed null model-based statistical testing and applied FDR correction (Benjamini-Hochberg) to account for multiple hypothesis testing.

## Whole-genome bisulfite sequencing (WGBS), data process, and joint analysis with bulk RNA-seq

WGBS libraries were prepared using the tPBAT protocol, as described by Miura et al.[99,100]. Briefly, 100 ng of genomic DNA was spiked with 1% (w/w) unmethylated lambda DNA (Promega) as a control for bisulfite conversion efficiency. The libraries were sequenced by Macrogen Japan Inc. on the Illumina HiSeq X Ten system. Raw sequencing reads were mapped to the hg38 reference genome using BMap (v1.0)[100], and methylation levels for each cytosine were calculated only when covered by a minimum of ten reads to ensure reliability.

Processed methylation data were analyzed using the R package methylKit (v1.20.0)[101] to quantify methylation levels at individual CpG sites. Differential methylation in promoter regions (defined as −1200 bp to +300 bp relative to the transcription start site, TSS) was identified based on two criteria: (1) a methylation difference (meth.diff) greater than 30%, and (2) a statistically significant q-value of less than 0.05.

To investigate the relationship between DNA methylation and gene expression, WGBS data were integrated with bulk RNA-seq data. Promoter methylation differences (meth.diff) were correlated with gene expression changes (log2FC) using Pearson correlation. Genes with missing values (≤7 in either dataset) were excluded, resulting in 15,644 genes for downstream analysis. Strongly anti-correlated genes (correlation coefficient < −0.5, $p < 0.05$) were retained (1896 genes), and those potentially influenced by auxin (meth.diff > 30%, adj. $p < 0.05$) were removed, yielding a final set of 1695 genes. Functional enrichment analysis was performed using the clusterProfiler package (v4.6.2) in R to identify biological pathways associated with methylation-mediated gene regulation. MSigDB Hallmark gene sets were employed to characterize the functional landscape of the 1695 correlated genes.

## Single cell RNA (scRNA) library preparation and sequencing

Single-cell capture and library preparation were performed using the Chromium GEM-X Single Cell 3' Kit v4 (1000686, 4 reactions; 10x Genomics) according to the manufacturer's protocol. Briefly, cells were partitioned into Gel Bead-In-EMulsions (GEMs) using the Chromium controller (10x Genomics). Reverse transcription was carried out within GEMs at 53 °C for 45 min, followed by enzyme inactivation at 85 °C for 5 min. Post-reaction cleanup was performed using DynaBeads MyOne Silane Beads (Thermo Fisher Scientific). The resulting cDNA was amplified using the following thermal cycling protocol: initial denaturation at 98 °C for 3 min, followed by 11 cycles of denaturation (98 °C for 15 s), annealing (67 °C for 20 s), and extension (72 °C for 1 min), with a final extension at 72 °C for 1 min. Amplified cDNA was purified using the SPRIselect Reagent Kit (Beckman Coulter), and fragment size distribution was assessed using the High Sensitivity DNA Kit on a 2100 Bioanalyzer (Agilent Technologies).

Sequencing libraries were constructed through enzymatic fragmentation, end repair, A-tailing, adapter ligation, and sample index PCR, with SPRIselect beads (Beckman Coulter) used for cleanup between each step. Final libraries were size-selected, quantified using the Qubit dsDNA High Sensitivity Assay (Thermo Fisher Scientific), and their fragment size distribution was verified using the High Sensitivity DNA Kit on the 2100 Bioanalyzer.

Libraries were pooled at equimolar concentrations and sequenced on an Illumina NextSeq 2000 platform using P3 100-cycle chemistry. The read configuration consisted of 28 bp for Read 1 (10x barcode + UMI), 10 bp each for the i7 and i5 indices, and 90 bp for Read 2 (transcript).

## scRNA sequencing data processing

Raw sequencing data (FASTQ files) were processed using the Cell Ranger pipeline (version 8.0, 10x Genomics) for demultiplexing. Sample-specific cell barcodes and unique molecular identifiers (UMIs) were extracted. Reads were aligned to the GRCh38 reference genome using Cell Ranger count (10x Genomics). Gene expression matrices (counts per cell per gene) were generated, retaining only high-quality cells with greater than 200 genes or <20% mitochondrial reads. Downstream analysis was performed using Scanpy (version 1.11.0) tool[102]. Data normalization and dimensionality reduction were performed, followed by clustering using the Leiden algorithm at a resolution of 0.80. Cell populations were visualized via uniform manifold approximation and projection (UMAP).

Large-scale chromosomal alterations were inferred from scRNA-seq data using inferCNV (v1.22.0). Non-malignant microglia/macrophages and oligodendrocytes served as diploid reference populations, establishing baseline gene expression levels. Target analysis included three experimental groups: (1) HCT116 cell lines (WT, U$^{AID1}$, D$^{AID1}$, and UD$^{AID1}$) with 8 days of Dox/Auxin treatment, (2) positive control samples consisting of six patient-derived aneuploid glioma lines (MGH36, MGH60, MGH53, MGH54, MGH93, and MGH97) from Massachusetts General Hospital, and (3) negative diploid controls. The data of the glioma lines and diploid controls were downloaded from a publicly available dataset hosted by the Broad Institute (https://github.com/broadinstitute/inferCNV_examples). Protein-coding genes were ordered by genomic position (GRCh38 assembly) and analyzed in 100-gene sliding windows to improve signal detection. A hidden Markov model (HMM) was applied to distinguish true CNVs from technical noise, with chromosomal gains and losses defined as regions showing consistent log2 expression ratios >0.1 or <−0.1, respectively, relative to diploid references. Results were visualized as heatmaps with genomic coordinates along the $x$-axis and single cells on the $y$-axis, using a red-blue color scale to represent relative copy number gains and losses.

## Statistics and reproducibility

All quantitative experiments were statistically analyzed using Graph-Pad Prism (v10.3.1, GraphPad Software Inc., San Diego, CA) and presented as mean ± SD or median. Comparisons between more than two groups were assessed by one-way ANOVA, two-way ANOVA, or Kruskal−Wallis test, followed by multiple comparisons test. Statistical details of each experiment can be found in the respective figure

legends. Principal component analysis (PCA), differentially expressed gene analysis, hierarchical clustering analysis, and heatmaps were generated by R software (v4.1.0). The exact p-values were displayed on the graphs when needed. $p < 0.05$ was considered significant. ns, not significant; $*p < 0.05$; $**p < 0.01$; $***p < 0.001$; $****p < 0.0001$ for indicated comparisons. All data were repeated independently to ensure reproducibility, with the number of replicates (N) specified for each experiment in the figure legends. Data that include three biological replicates, each from at least two independent experiments with similar results. Where representative images are shown, experiments were repeated at least three times unless otherwise noted.

### Reporting summary
Further information on research design is available in the Nature Portfolio Reporting Summary linked to this article.

### Data availability
The bulk RNA-seq data used and generated in this study have been deposited in the Gene Expression Omnibus (GEO) database under accession code GSE249536 and GSE278103, respectively. The WGBS data used and generated in this study are available in the GEO database under accession code GSE236026 and GSE278681, respectively. The processed scRNA-seq data are available at Zenodo (https://doi.org/10.5281/zenodo.15105182). Source data are provided with this paper.

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

## Acknowledgements

This work was supported by the following grants: Agence Nationale de la Recherche (ANR-23-CE12-0015-01, to PAD, GC, and DB), Institut National du Cancer (INCA_18350, to PAD), and Fondation pour la Recherche Médicale (EQU202503019988). The team of NM is supported by Fondation pour la Recherche Médicale (EQU202103012774), Agence Nationale pour la Recherche (ANR-21-CE15-0037-01 and ANR-21-CE13-0030-02), Fondation ARC (ILUMINAGE), Sidaction (22-2-AEQ-13411), and the Clayco Foundation. XC received support from "Fondation pour la Recherche Médicale" for the last year of her PhD (FDT202404018544). KY was supported by ARC foundation CDD Post-doctorant en France (Ref. No. PDF20181208337), Labex "Who Am I?" postdoctoral support, and a JSPS Postdoctoral Fellowship for Research Abroad (Ref. No. 202260432). The work of FM and TI was supported by Research Support Project for Life Science and Drug Discovery (Basis for Supporting Innovative Drug Discovery and Life Science Research (BINDS)) from AMED under Grant Number JP23ama121022. The authors are grateful to the ImagoSeine core facility of Institut Jacques Monod, member of France-BioImaging (ANR-10-INBS-04), with the support of Plan Cancer, Region Ile-de-France, and Fondation Bettencourt Schueller (R03/75-79). We also thank the Vectorology platform, Epigenetics platform, Microscopy platform, and Bioinformatics/Biostatistics Core Facility (BIBS) at the CNRS Epigenetics and Cell Fate Unit (Université Paris Cité) for providing access and technical advice. We are especially grateful to the Hist'IM facility (Institut Cochin) for excellent histology support. We are very indebted to Saadi Khochbin, Allison Bardin, Corinne Abadie, Bill Keyes, Julien Cherfils-Vicini, Vjekoslav Dulic, Clément Hua, Nadine Laguette, Han Li, Raphael Margueron, François Radvanyi, Laurent Reber, and David Roulois for useful advice. We thank the following colleagues for the gift of useful reagents: Aurélien Doucet, Marc-Henri Stern, Fadila Rayah, Catherine Postic, Emma Lamana, and Pierre Bruhns.

## Author contributions

X.C., K.Y., and P.A.D. conceived, designed, and supervised the project. X.C., K.Y., and B.R. established the constructs and cell lines used in this study. X.C., K.Y., and B.R. performed molecular and cellular biology experiments, collected and analyzed data, with the assistance of M.N., L.F., and S.L. X.C. analyzed, interpreted, and visualized the bulk RNA-seq data and single cell RNA-seq data. K.Y. and X.C. analyzed, interpreted, and visualized the WGBS data. D.G. and D.V. performed xenograft experiments, with the assistance of N. Martin. X.L. assisted with experiments related to cGAS and STING. X.C. and P.D. performed histology analyses. F.M. performed experiments related to DNA methylation. X.C. and B.R. wrote the original draft of the manuscript. P.A.D. reviewed and edited the draft. T.I., N. Manel, G.C., M.K., D.B., and P.A.D. acquired funding.

## Competing interests

The authors declare no competing interests.
