## [Peer Review file · Nature Communications]

DNA methylation protects cancer cells against senescence

Corresponding Author: Dr Pierre Antoine Defossez

Version 0:

Reviewer comments:

Reviewer #1

(Remarks to the Author)

The most widely studied and commonly used chemical agent to inhibit DNA methylation is 5-azadeoxycytidine. 5-azadC gets incorporated into DNA, is crosslinked to DNMT proteins, and elicits a DNA damage response in addition to promoting loss of DNA methylation. In the present study, the authors developed cellular systems from (mostly) colorectal cancer lines to investigate the consequences of DNA demethylation when it occurs in the absence of DNA damage. They used degron-based elimination of the DNA methyltransferase DNMT1, of the DNMT1 cooperative protein UHRF1, and targeted the combination of the two simultaneously. A noncovalent DNMT1 inhibitor that does not get incorporated into DNA was used as well. The main observation from this work is that the loss of the methylation maintenance proteins causes cellular senescence as shown by several types of standard assays. Additional follow-up work was done to understand the potential mechanisms of this novel senescence pathway. The senescence phenotype was independent of p53 and Rb, involved cytoplasmic p21, which inhibited apoptosis, and cGAS, which played a STING-independent role in the nucleus and not in the cytoplasm. The experiments are generally well designed and executed. Statistical analysis is provided. Although the authors managed to exclude several possible mechanisms that are often linked to cellular senescence in other systems, they did not identify the pathway leading from DNA demethylation to senescence. The manuscript has several other shortcomings.

1) The extent of DNA demethylation that is predicted to occur in the degron systems was not clear. There is only one piece of data in the last Figure (Fig. S6B) although this important piece of information should be introduced at the very beginning. The LUMA assay they used for measuring total methylation at restriction sites is quite limited. It does not allow one to determine which genomic sequences are becoming hypomethylated and to what extent.

2) One commonly observed consequence of treating cells with DNA methylation inhibitors is the upregulation of certain methylated transposable elements (TE) and other repetitive DNA sequences, which may elicit an immune response. However, the authors did not investigate if this is happening in their system and how such TE activation would contribute to the senescence phenotypes observed.

3) Even though the authors were able to exclude the involvement of otherwise known pro-senescence mechanisms (e.g., p53, Rb), they have not actually arrived at a conclusion as to what it is that leads from DNA demethylation to senescence.

4) The data color scheme in Figure 4A is not very informative.

5) I don't understand Figure S4. In panel A, p21 is in the cytoplasm but in panel B it seems to be in the nucleus. There is another band near 16 kDa in the p21 blot of panel A. What is this band? The same issues may be raised about panels E and F of main Figure 4.

6) Figure 5:

Do the authors have any explanation as to why only the dual degradation of UHRF1 and DNMT1 leads to upregulation of cGAS even though the genome is expected to be demethylated in the single degradation systems as well.

Reviewer #2

(Remarks to the Author)

Using the auxin-degron system, Chen et al show evidence that depletion of DNMT1 and/or UHRF1 causes senescence in cancer cells without substantial DDR. The phenotype appears independent of p53 and p16/RB, based on the use of RNAi and/or viral oncoproteins, E6/E7. They show that cytoplasmic p21 is upregulated, inhibiting apoptosis, during senescence caused by depletion of DNMT1 and/or UHRF1. Finally, they show cGAS upregulation at nuclei. This cGAS contributes to the SASP, but it is sting-independent.

Decoupling loss of DNA methylation and DNA damage response (DDR) is an important question, and overall, the experiments are well conducted. However, some data interpretation appears too simplistic.

Generally, since this is a new type of senescence, it is critical to show a sustained cell-cycle arrest. For example, to assess long-term arrest and heterogeneity, a colony-formation assay (w/o the ligands) would be helpful.

Fig. 2C, UD-AID1 day 0: why there are so many (142) DEGs? Is this due to the system's leakiness?

Fig. 3A, B (DNA damage response): The positive control used is an acute phase of HU/Eto treatment (4h). This is fine for showing cells' acute DDR capacity, but senescence is typically associated with 'persistent DDR', which is quite modest. To show a lack of DDR in the degron cells, persistent DDR (typical senescence) would be necessary as another control.

Fig. S3F: E6-mediated p53 degradation is efficient under unstressed conditions, but p53 can be readily stabilized in the presence of E6 under certain conditions (e.g. DNA damage). To conclude that p53 is not involved in this type of senescence, they need to assess p53 levels in the presence of dox/auxin. Also, I wonder what happens to p21 when E6 is expressed.

Senescence, particularly in human cells, is highly complex, with multiple effectors compensating for each other to maintain the senescence arrest. Thus, it is too simplistic to exclude p53/Rb involvement based on the E6/7 data alone. For example, in Fig. 3H, I wonder if it is possible that E6/7 partially rescues cell cycle arrest but promotes cell death (E7 can sensitize cells to cell death).

Fig. 3J: Along the same lines, it would be helpful to show other senescence-related gene expression, including E2F-targets. I would expect E2F-targets to be upregulated in the presence of E6/7 but then, why cells do not proliferate (Fig. 3H). Again are cells more apoptotic? Or a decoupling between proliferation and cell-cycle gene expression?

Fig. 6G: SA-b-gal may not be the best marker in this case since, in Fig. 6I, they show a significant accumulation of macrophages, in which lysosomal activity is high. How about p21 staining?

Other points:

Fig. 1K: they need to confirm cell cycle arrest.

Fig. 2: in d0 UD-AID1 cells show substantial DE genes. Is this due to a leakiness?

Fig. S1, 4C: since Hoechst is membrane permeable, it doesn't distinguish between "early" apoptosis and secondary necrosis?

Reviewer #3

(Remarks to the Author)

The manuscript presents new findings that contribute to our understanding of the relationship between DNA methylation and cellular senescence in cancer cells. The use of degron alleles to dissect these effects is particularly innovative, as it addresses limitations in previous studies involving broad-spectrum inhibitors like 5-aza-deoxycytidine. Moreover, the identification of alternative pathways leading to senescence broadens the scope for potential therapeutic interventions. Overall, this work appears to offer intriguing insights into the intrinsic effects of DNA methylation loss on cancer cells. However, quite a few key questions need to be addressed:

While the advantages of the degron system are articulated, it is crucial to integrate insights from prior loss-of-function studies. A comparison with existing research, such as investigations involving DNMT conditional knockout models, could provide valuable context and facilitate a more comprehensive understanding of the observed senescence and interferon responses.

In Fig 1 depicting evaluations of senescence following DNMT1 and UHRF1 depletion, discrepancies are noted in the trends observed. For example, in Fig 1H, at Day 4 after depletion of DNMT1 alone shows no increase of SA-b-gal %area, while UHRF1 does, but in Fig 1F, at the same time DNMT1 depletion already caused nuclear area increase. There are other discordance between different panels of Fig 1. The authors should discuss about these differences.

One noticeable gap in the current study is the limited experimental evaluation of DNA methylation changes, except for one LUMA assay on the global loss. To establish a more robust causal relationship between DNA methylation alterations and specific gene upregulation, a genome-wide analysis is deemed essential. Ideally, whole-genome bisulfite sequencing at selected time points for the degron lines or comparable conditions should be integrated with corresponding RNA-seq. Alternatively, leveraging related published datasets could serve as a surrogate to validate and elaborate on observed gene expression changes.

Along this line, the manuscript would benefit from a deeper exploration of the mechanisms underlying the observed increase in p21 expression. Understanding the specific genomic regions affected by DNA methylation changes leading to p21 upregulation, or identifying the upstream triggers of this phenomenon, would enrich the interpretation of the findings and

elucidate the molecular underpinnings of senescence induction.

Furthermore, the correlation between nuclear cGAS and SASP expression raises intriguing questions regarding the relationship between DNA hypomethylation and cGAS-mediated effects on senescence. Elaborating on the mechanisms linking DNA hypomethylation with the nuclear role of cGAS and its impact on target gene expression could provide new insights. Similar here, WGBS or methylation-sensitive PCR on regions respond to DNMT1/UHRF1 depletion would significantly enhance the manuscript's coherence and relevance to hypomethylation.

Regarding the therapeutic implications highlighted in the study, I am not sure what to make of the discrepancy of Fig 5D and 5E, does that mean reduced level of senescence was not reflect in cellular growth? Please explain.

Minor points:

At Page 8, one reference to Fig S1D should be to Fig S2D.

Fig 6E showed a much stronger reduction of Ki67 signal when UHRF1 is deprived than is DNMT1, but the quantification in Fig 6F showed the opposite. Why?

Reviewer #4

(Remarks to the Author)

Chen et al. found that induction of global hypomethylation by the depletion of UHRF1 and/or DNMT1 proteins from HCT116 cells resulted in cellular senescence but not apoptosis. They further demonstrated that various types of cancer cells enter senescence upon the administration of a DNMT1 inhibitor. The authors' work represents the first report of hypomethylation-induced senescence in "cancer cells". Through well-designed and careful experiments, they demonstrated that this induced senescence is independent of p53 and Rb but involves p21 and cGAS. The authors also demonstrated that tumor cells can also be made senescent in vivo by decreasing DNA methylation.

Although the authors made good use of their HCT116 cells with degron alleles to figure out which ones of known molecular pathways that underpin senescence are active in the demethylation-induced senescence of cancer cells, the weakness of this work is the lack of information through what mechanisms DNA hypomethylation trigger senescence in cancer cells. Many of the nonlethal stresses have been shown to induce cellular senescence as a stress response (PMID: 25312810), and DNA hypomethylation may be merely another example of the nonlethal stresses. To address this concern and improve the novelty and originality of this work, the authors need to clarify how DNA methylation directly prevents cancer cells from cellular senescence and how DNA hypomethylation triggers senescence pathways in cancer cells.

Major comments:

1. This reviewer thinks that the authors need to pursuit possible mechanisms through which DNA demethylation leads to senescence of cancer cells. In their previous work (Yamaguchi K et al. 2024, PMID: 38580649), the global DNA methylation level of HCT116 cells is shown to be dropped from 70% to 20-40% upon the depletion of DNMT1 and/or UHRF1.

Previous studies by other groups have demonstrated that such hypomethylation changes DNA replication timing and 3D genome organization in HCT116 cells (Qu Q et al, 2021, PMID: 34551299) and causes chromosomal instability possibly originated from centromeric hypomethylation in hTERT RPE1 cells (Besselink N et al, 2023, PMID: 37106015). Comparison of 3D genome organization in the authors' HCT116 cells with degron alleles upon DNA demethylation and that in replicative senescence may help narrow down genomic regions contributing to hypomethylation-induced senescence in cancer cells. Because aneuploidy has been proposed as a key trigger into senescence (Macedo JC et al. 2018, PMID: 30026603; Kirsch-Volders M, Fenech M. 2023, PMID: 37866738), it is important to determine the frequency of aneuploidy upon induction of hypomethylation in the authors' HCT116 derivative cells.

Induction of genome-wide demethylation in cancer cells may lead to reactivation of retrotransposable elements and loss of heterochromatin, the former of which has been shown to drive IFN response in senescent fibroblasts (De Cecco M et al, 2019, PMID: 30728521) and the latter of which has been shown to be implicated in replicative senescence (Zhang X et al. 2021, PMID: 34140314; Deng L et al. 2019, PMID: 31350386). By examining the extent of each of the events listed here in the authors' HCT116 derivative cells upon the induction of demethylation, the authors will be able to narrow down possible mechanisms of demethylation induced senescence in cancer cells.

The promoters of cGAS and STING are reported to be highly methylated and their expression is repressed in cancer cells (PMID: 29367762). Rather than genome-wide changes triggered by DNA demethylation, can the demethylation-induced senescence in cancer cells be explained by the de-repression of a limited number of such senescent pathway genes? Such a possibility should also be assessed and discussed in this paper.

2. The following two papers are highly relevant to the authors' present work and should be appropriately cited and discussed in the authors' manuscript. Xie et al. (2017, PMID: 28277545) demonstrated that DNMT1 knockdown in young human skin fibroblasts induced senescence phenotype. Young et al. (2017, PMID: 28394343) demonstrated that UHRF1 knockdown in human fibroblasts (2BC and IMR90 lines) induced senescence.

Reviewer #5

(Remarks to the Author)

Version 1:

Reviewer comments:

Reviewer #1

(Remarks to the Author)

The authors have made major efforts to address the comments of the four reviewers. They added 22 new figure panels to the already existing one hundred or so. The new data have significantly improved the manuscript. I am satisfied with the revisions.

Reviewer #2

(Remarks to the Author)

The authors have addressed my questions satisfactorily. I have no further comments.

Reviewer #3

(Remarks to the Author)

The authors have fully addressed my concerns. The revised manuscript is significantly improved with the addition of substantial new data and analyses that rule out alternatives, and strengthen the main conclusions. While the ultimate upstream molecular sensor that links DNA hypomethylation to the downstream effectors (like p21 and cGAS activation) remains to be identified, the current study provides a thorough characterization of this novel, non-canonical senescence pathway in cancer cells and its key features.

Reviewer #4

(Remarks to the Author)

The authors satisfactorily addressed my comments. I found the reasons the authors mentioned as to why they did not examine changes in 3D genomic organization were reasonable. I hope the authors clarify the causal mechanisms of induced senescence by DNA demethylation in cancer cells through additional efforts such as genetic screens and thorough epigenetic profiling in the future.

Reviewer #5

(Remarks to the Author)

Rebuttal letter

We are grateful to the five reviewers for their insightful comments; they have been duly taken account and have helped us improve the manuscript. The comments are copied verbatim below in black, with our responses in blue.

The other points were addressed experimentally, resulting in **22 new panels** (new Figures 1I-J, 1M, S1J, 2E-G, S2I-J, S3F-L, S4C-D, S5A-B, S5D-E, S6G-I), **addition of data in 3 existing panels** (Figs 1L, 3B, and S3E) and lastly **2 figures for the reviewers** that will not appear in the final MS.

The experiments we have added include:

- proliferation and colony-forming assays reinforcing the notion that inhibition of DNA methylation causes a proliferation arrest (Figs 1I, J, M; S1J)
- demonstration that DNMT1 inhibition causes senescence in 3 new cell lines, further ruling out a requirement for p53 or Rb in senescence induced by loss of DNA methylation (Figs 1L-M, S3F)
- deep WGBS (3 degron lines at days 6 and 8, 18 samples), integration of RNA-seq and WGBS data determining which part of the transcriptional response directly results from loss of DNA methylation (Figs 2G, S4C-D, S5A-B)
- single-cell RNA-seq confirming and refining the senescence signature seen in bulk RNA-seq (Figs 2E-F)
- analysis of TE expression (Fig S2J)
- experiments with persistent DDR activation (Fig 3B)
- additional IHC showing that tumors with decreased DNA methylation express p21 and that the SA- β -gal staining is not caused by macrophages (Figs S6G-I)
- expression studies and shRNA experiments showing that L1 repeats are not involved in the senescent phenotype (Figs S3H-L)
- single-cells analyses establishing that the senescent cells are not aneuploid (Fig S3M)
- gain-of-function experiments on cGAS (Fig S5E)

All the new experiments confirm and extend our original findings. These points having been addressed, we hope that you'll find this revised version of the manuscript suitable for publication in Nature Communications.

Reviewer #1 (Remarks to the Author):

The most widely studied and commonly used chemical agent to inhibit DNA methylation is 5-azadeoxycytidine. 5-azadC gets incorporated into DNA, is crosslinked to DNMT proteins, and elicits a DNA damage response in addition to promoting loss of DNA methylation. In the present study, the authors developed cellular systems from (mostly) colorectal cancer lines to investigate the consequences of DNA demethylation when it occurs in the absence of DNA damage. They used degron-based elimination of the DNA methyltransferase DNMT1, of the DNMT1 cooperative protein UHRF1, and targeted the combination of the two simultaneously. A noncovalent DNMT1 inhibitor that does not get incorporated into DNA was used as well. The main observation from this work is that the loss of the methylation maintenance proteins causes cellular senescence as shown by several types of standard assays. Additional follow-up work was done to understand the potential mechanisms of this novel senescence pathway. The senescence phenotype was independent of p53 and Rb, involved cytoplasmic p21, which inhibited apoptosis, and cGAS, which played a STING-independent role in the nucleus and not in the cytoplasm.

> We thank you for this thorough and accurate summary of the manuscript.

The experiments are generally well designed and executed. Statistical analysis is provided.

> We are grateful for your positive assessment of our work.

Although the authors managed to exclude several possible mechanisms that are often linked to cellular senescence in other systems, they did not identify the pathway leading from DNA demethylation to senescence.

> Thank you for pointing this out. This point is discussed in detail in answer to your point 3, below (page 5 of the current document)

The manuscript has several other shortcomings.

> Thank you for your expert advice. All the points you raised have been addressed, as detailed below. We are grateful for this opportunity to make our paper stronger.

1) The extent of DNA demethylation that is predicted to occur in the degron systems was not clear. There is only one piece of data in the last Figure (Fig. S6B) although this important piece of information should be introduced at the very beginning. The LUMA assay they used for measuring total methylation at restriction sites is quite limited. It does not allow one to determine which genomic sequences are becoming hypomethylated and to what extent.

> Thank you for raising this key point. We agree that the limited experimental evaluation of DNA methylation changes was a gap of the initial study. Our previous publication (Yamaguchi 2024) supplied some of the data, specifically deep WGBS at days 0, 2, and 4. For this revision, we performed deep WGBS at days 6 and 8, and reanalyzed the whole set of data. Here is a summary of the new data provided in the revision:

	Day 0	Day 2	Day 4	Day 6	Day 8
Deep WGBS	Yamaguchi 2024	Yamaguchi 2024	Yamaguchi 2024	Done for the revision	Done for the revision

The WGBS data is completely consistent with the data previously reported in Yamaguchi 2024 with LUMA, LC-MS, and shallow WGBS. Because the data are so similar, we have opted not to show them again as a figure in the manuscript. But here is a piece of data from Yamaguchi 2024 that you may find informative:

Reviewer panel: kinetics of DNA methylation loss, determined by shallow WGBS, from Yamaguchi 2024

2) One commonly observed consequence of treating cells with DNA methylation inhibitors is the upregulation of certain methylated transposable elements (TE) and other repetitive DNA sequences, which may elicit an immune response. However, the authors did not investigate if this is happening in their system and how such TE activation would contribute to the senescence phenotypes observed.

> Thank you for pointing this out. To fill in the gap you have identified, we examined TE expression in our system, and also performed functional experiments.

1-Expression of TE

We reanalyzed our bulk RNA-seq data with the Tetranscripts package (PMID: 26206304). The data are shown in the new panel S2I, copied below

New panel S2I: analysis of TE expression in the bulk RNA-seq data

Among the TEs that are induced, we found a modest induction of L1 transposons at certain timepoints. Given the known role of L1 in replicative senescence, we examined L1 expression more specifically, at the protein level, using an antibody against ORF1p. This revealed an induction of L1 ORF1p in UHRF1/DNMT1 degnon after 8 days of depletion (New panel S3H, copied below).

New panel S3H: detection of L1 ORF1p in senescent cells.

2-Functional experiments on L1s

For this we performed shRNA against L1, using a vector published by Dr Cristofari (Philippe ELife 2016, PMID: 27016617). We infected and selected the degron cells, then treated them with 5-aza-C to induce ORF1p to monitor the efficiency of the shRNA. This experiment now appears as panel S3I in the paper, copied below for your convenience:

New panel S3I: validation of shORF1 vector

The conclusion from this experiment is that the shRNA vector works, as previously published. Next, we asked whether stably knocking down L1s had an effect on proliferation or senescence in our cells. As shown in the new panel S3J, copied below, the shORF1 vector did not rescue the growth defect in any of our cell lines.

New panel S3J: no rescue of growth defect by shORF1 vector

Besides the growth phenotype, we also assessed whether shORF1 remediated SA- β -gal staining or SASP expression, and found that it did not (New panels S3K-L, below).

New panels S3K-L: no rescue of SA- β -gal staining or SASP expression by shORF1 vector

To conclude and answer your question, these experiments rule out a role for L1 elements in the senescence we observe.

3) Even though the authors were able to exclude the involvement of otherwise known pro-senescence mechanisms (e.g., p53, Rb), they have not actually arrived at a conclusion as to what it is that leads from DNA demethylation to senescence.

> Your point is well taken, thank you. As you say, we have firmly excluded known pro-senescence mechanisms (p53, Rb). This in itself is important conceptually —it differentiates our observations from other cases of senescence—, and also practically —it means that senescence induced by loss of DNA methylation may be an avenue to inhibit the growth of tumors lacking p53 and/or Rb.

We did our utmost to identify the effectors of senescence in our systems or, as you rightly say, to “identify what it is that leads from DNA demethylation to senescence”. During the revision process, we tested 2 more candidates: expression of L1s and aneuploidy. However, we conclusively show that none of these causes senescence in our system.

As you doubtlessly appreciate, identifying the causal mechanisms is an open-ended pursuit, and there is no telling how long it might take, especially since we need to pivot from a candidate approach to an unbiased one. We are deeply interested in this question for the future, and for example we are thinking about setting up genetic screens to try to arrive at an answer. However, again, we feel this has to be reserved for another paper, a few years from now.

The initial study we submitted, and even more so its revised version, extensively characterizes a new phenomenon, which is of interest to a broad community, and that relies on a mechanism that we show is non-canonical. This seems to us to be on par with articles published by Nature Communications, and we hope you will agree with this assessment.

4) The data color scheme in Figure 4A is not very informative.

> Thank you for helping us improve this panel. We have modified it as you requested.

5) I don't understand Figure S4. In panel A, p21 is in the cytoplasm but in panel B it seems to be in the nucleus.

> Thank you for raising this point, and we apologize for the confusion. We now realize that panels S4A and S4B look very similar. Therefore, one may conclude that they represent the same thing, i.e. the results of a fractionation assay to determine whether p21 is cytoplasmic or nuclear. However, this is not the case: while panel S4A indeed reports a fractionation assay, panel S5B validates the efficiency of the p21 shRNA. We have modified the figures to make this clearer. Here is the new version:

Modified panels S4A and S4B. Titles modified to increase clarity

There is another band near 16 kDa in the p21 blot of panel A. What is this band? The same issues may be raised about panels E and F of main Figure 4.

> Thank you for this comment. The band near 16 kDa that you pointed out in Panel A was non-specific. We have redone the western blots of panels 4E, 4F and S4A with a new antibody. Here are the updated 4E and 4F panels, as an illustration.

Modified panels 4E and 4F. The p21 western blot were redone.

6) Figure 5: Do the authors have any explanation as to why only the dual degradation of UHRF1 and DNMT1 leads to upregulation of cGAS even though the genome is expected to be demethylated in the single degradation systems as well.

> This is a good point, thank you. We believe the reason is the following : the dual degradation of UHRF1 and DNMT1 leads to steeper loss of DNA methylation than either single degradation. Our new panels S5A and S5B now show that the induction of cGAS is proportional to the degree of DNA demethylation, and is therefore highest in the dual degradation system.

New panels S5A-B. The induction of cGAS correlates with loss of DNA methylation at its promoter, which is greatest in the dual UD line.

Reviewer #2 (Remarks to the Author):

Using the auxin-degron system, Chen et al show evidence that depletion of DNMT1 and/or UHRF1 causes senescence in cancer cells without substantial DDR. The phenotype appears independent of p53 and p16/RB, based on the use of RNAi and/or viral oncoproteins, E6/E7. They show that cytoplasmic p21 is upregulated, inhibiting apoptosis, during senescence caused by depletion of DNMT1 and/or UHRF1. Finally, they show cGAS upregulation at nuclei. This cGAS contributes to the SASP, but it is sting-independent.

> Thank you for this precise summary of the manuscript. We appreciate the time you took in evaluating it.

Decoupling loss of DNA methylation and DNA damage response (DDR) is an important question, and overall, the experiments are well conducted.

> We agree with this point and we appreciate your positive comments on our work.

However, some data interpretation appears too simplistic.

Generally, since this is a new type of senescence, it is critical to show a sustained cell-cycle arrest. For example, to assess long-term arrest and heterogeneity, a colony-formation assay (w/o the ligands) would be helpful.

> Thank you for raising the important questions of long-term arrest and heterogeneity. We have performed 3 new sets of experiments to address these questions of yours.

1) is there really an arrest, or are we observing slow-cycling cells?

For this we exposed the cells to EdU for an extended period (18 hours, equivalent to one whole cell cycle in untreated cells). Under these conditions, all control cells incorporated EdU, as expected. In contrast, 40 to 60% of the cells depleted of UHRF1, DNMT1, or both failed to incorporate any EdU, suggesting they are either quiescent or senescent.

New panel 1I. A long EdU pulse demonstrates cell cycle arrest

2) is the arrest irreversible?

As you suggested, we performed a colony-formation assay (w/o the ligands). These data now constitute Panel 1J, copied below.

New panel 1J. A colony-formation assay demonstrates irreversible growth arrest.

There are 3 take-aways from these experiments. The first is that some cells are arrested long-term. This is especially obvious in the double-degrogen cells (UD^{AID1}), which hardly form any colonies, even in the absence of ligands. The second is that the decrease of colony-forming ability tracks with the degree of DNA methylation decrease (DNMT1 depletion is less severe than UHRF1 depletion, which is less severe than the double depletion). The third take-away is that the response is heterogeneous, as not all cells are irreversibly arrested.

3) What can we learn about the heterogeneity?

For this we performed single-cell RNA-seq (New panels 2E-F). The experiment was illuminating, revealing that some cells turn off the proliferation signature without expressing the senescence signature (clusters 5 and 6), whereas some others stop proliferating and express the senescence markers (cluster 7).

New panels 2E-F. Single-cell RNAseq allows the deconvolution of bulk RNA-seq data

Fig. 2C, UD-AID1 day 0: why there are so many (142) DEGs? Is this due to the system's leakiness?
 > You are right to point this out. We believe the explanation is not leakiness: in our previous paper (Yamaguchi Nat Comm 2024), we carefully ascertained, by western blot but also by Mass Spec, that the amounts of UHRF1-AID and DNMT1-AID were the same in all lines before the addition of auxin. Here is an illustration from that previous paper:

Figure 1C from Yamaguchi et al, 2024. Amount of AID-tagged protein in the indicated cells.

Instead, we think the differences between single degron lines (UHRF1-AID, DNMT1-AID), and the double degron line (UD-AID1) has to do with their history. Indeed, to make the double degron line, we have to start from a single degron line, as we illustrated in our previous paper:

Figure 1B from Yamaguchi et al, 2024. Procedure to generate the double degron line.

Based on the time it takes to generate and validate degron clones, we estimate the double degron line has gone through ~20-25 more population doubling than the single degron lines. During that time, some epigenetic drift has taken place, and we believe that explains why there are differentially expressed genes even in the absence of auxin.

Fig. 3A, B (DNA damage response): The positive control used is an acute phase of HU/Eto treatment (4h). This is fine for showing cells' acute DDR capacity, but senescence is typically associated with 'persistent DDR', which is quite modest. To show a lack of DDR in the degron cells, persistent DDR (typical senescence) would be necessary as another control.

> This is a good point, thank you. As per your suggestion, we redid the experiment, this time including cells having a 'persistent DDR'. For this, we rendered HCT116 cells with a 24-hour Etoposide treatment. We then washed away the medium and let the cells recover for 5 days. We made sure they were senescent by the SA-b-gal assay, as shown below.

New reviewer panel: cells with DNA-damage induced senescence, used as controls for persistent DDR in panel 3B.

We then used these cells as controls in the updated western blot of Figure 3B. They display chronic p53 stabilization, as well as presence of phospho-Chk2 and γ H2AX. On the same membrane, cells rendered senescent by decreased DNA methylation are negative for all these markers. This further supports the notion that senescence induced by loss of DNA methylation is independent of DNA damage.

Modified panel 3B. Cells with persistent DDR activation included as controls.

Fig. S3F: E6-mediated p53 degradation is efficient under unstressed conditions, but p53 can be readily stabilized in the presence of E6 under certain conditions (e.g. DNA damage). To conclude that p53 is not involved in this type of senescence, they need to assess p53 levels in the presence of dox/auxin. Also, I wonder what happens to p21 when E6 is expressed.

> We freely admit that we were unaware that E6-mediated p53 degradation could be less than total in certain conditions, such as DNA damage. Thank you for alerting us to this. It indeed makes it crucial to assess p53 levels in the presence of dox/auxin, which we have done as you suggested (New Figure S3F, copied below for your convenience).

New panel S3F: p53 is degraded by E6 even in the presence of auxin.

The levels of p53 in E6-expressing cells are low even in the presence of auxin. In addition, we have performed a number of experiments in cells mutant for p53, as explained below.

Senescence, particularly in human cells, is highly complex, with multiple effectors compensating for each other to maintain the senescence arrest. Thus, it is too simplistic to exclude p53/Rb involvement based on the E6/7 data alone. For example, in Fig. 3H, I wonder if it is possible that E6/7 partially rescues cell cycle arrest but promotes cell death (E7 can sensitize cells to cell death).

> Another good point on the danger of simplistically overinterpreting the E6/E7 data, thank you. If we may, we'd like to point out that not all of our conclusions derive from the E6/E7 experiments. In particular, in Figure 1, we had explored a variety of cell lines with varying p53/Rb status. Based on your recommendation, we sought to reinforce our conclusions regarding p53 and Rb. For this, we performed experiments with additional lines in which Rb is inactivated (MDA-MB-468 and Saos-2):

Cell line	Tumor type	p53 status	Rb status
HCT116	Colorectal adenocarcinoma	WT	WT
DLD1	Colorectal adenocarcinoma	Mutant (S241F)	WT
HT29	Colorectal adenocarcinoma	Mutant (R273H)	WT
A549	Non-small-cell lung adenocarcinoma	WT	WT
MCF7	Breast adenocarcinoma	WT	WT
HeLa	Cervical carcinoma	Inactivation by E6	Inactivation by E7
MDA-MB-468	Breast adenocarcinoma	Mutant (R273H)	Deletion (c.265_2787del12523)
Saos-2	Osteosarcoma	Deletion Ex4-Ex8	Deletion

All of these cells stain positive for SA- β -gal and stop incorporating EdU upon chronic exposure to the DNMT1 inhibitor GSK3685032

New panels 1L-M: Effect of DNMT1 inhibition on cells with various p53/Rb status

We also used the p53 KO derivatives of HCT116, and those become SA-β-gal positive after treatment with GSK3685032, further ruling out a requirement for p53 in this system (New panel S3F, copied below).

New panel S3F: HCT116 p53^{-/-} cells become SA-β-gal positive after DNMTi treatment

Lastly, we verified the protein expression of p53 and Rb in all cell lines (New panel S3G, copied below). The results were as expected, validating the idea that neither p53 nor Rb is required for DNA methylation loss to cause senescence in cancer cells.

New panel S3G: verification of p53/Rb expression in all cell lines

Fig. 3J: Along the same lines, it would be helpful to show other senescence-related gene expression, including E2F-targets. I would expect E2F-targets to be upregulated in the presence of E6/7 but then, why cells do not proliferate (Fig. 3H). Again are cells more apoptotic? Or a decoupling between proliferation and cell-cycle gene expression?

> Thank you for this perceptive question. You are right that caution should be exercised when interpreting the E6/E7 experiments. However, the new experiments on cells with mutations in p53 and/or Rb conclusively show that neither protein is required for senescence caused by loss of DNA methylation. Unless we are mistaken, this removes the necessity for further controls on E6/E7-expressing cells.

Fig. 6G: SA-b-gal may not be the best marker in this case since, in Fig. 6I, they show a significant accumulation of macrophages, in which lysosomal activity is high. How about p21 staining?

> Thank you for raising the important caveat that macrophages, in which lysosomal activity is high, can confound the interpretation of SA-b-gal staining, and for suggesting p21 as an alternative. We have followed both your suggestions as follows.

1-Is the SA-b-gal stain due to macrophages?

To test this, we performed IHC for the macrophage marker F4/80 on slides that had been previously stained for SA-b-gal. The two stains are clearly distinct (new panel S6G, below).

New panel S6G: SA-b-gal staining (blue) and IHC for the macrophage marker F4/80 (brown), on sections of auxin-treated tumors, counterstained with Nuclear Fast Red.

2-How about p21 staining?

As you recommended, we also performed p21 staining on the tumor sections. We detect positive staining in the auxin-treated tumors, supporting the notion that these tumors contain senescent cells (New panel S6H, reproduced below).

New panel S6H: IHC for p21 on sections of auxin-treated tumors

Other points:

Fig. 1K: they need to confirm cell cycle arrest.

> This has been done as you requested. We submitted the cells to a long EdU pulse (20 hours). In control conditions, virtually all the cells incorporate EdU in that time window. After exposure to the DNMT1 inhibitor, 40 to 70% of cells fail to incorporate EdU in the same period, establishing that they have indeed undergone cell cycle arrest. (New panels 1M and S1J).

New panel S1J: cells exposed to chronic DNMT1 inhibition fail to incorporate EdU during a 20-hour pulse

Fig. 2: in d0 UD-AID1 cells show substantial DE genes. Is this due to a leakiness?

> Thank you for this comment. We believe it is similar to the point you raised above: "Fig. 2C, UD-AID1 day 0: why there are so many (142) DEGs? Is this due to the system's leakiness?". We would therefore like to direct you to the answer we have made on page 9, thank you.

Fig. S1, 4C: since Hoechst is membrane permeable, it doesn't distinguish between "early" apoptosis and secondary necrosis?

> Yes you are right. To eliminate this problem, we re-analyzed all the FACS data after collapsing the Hoechst axis. Now we score only along the Annexin V axis. Both panels have been corrected, with new panel S1A shown below as an illustration. The conclusions are unaffected.

New panel S1A. Quantification of Annexin-positive cells.

Reviewer #3 (Remarks to the Author):

The manuscript presents new findings that contribute to our understanding of the relationship between DNA methylation and cellular senescence in cancer cells. The use of degron alleles to dissect these effects is particularly innovative, as it addresses limitations in previous studies involving broad-spectrum inhibitors like 5-aza-deoxycytidine. Moreover, the identification of alternative pathways leading to senescence broadens the scope for potential therapeutic interventions. Overall, this work appears to offer intriguing insights into the intrinsic effects of DNA methylation loss on cancer cells.

> We are very grateful for your positive assessment on the originality and importance of our work, thank you.

However, quite a few key questions needs to be addressed:

While the advantages of the degron system are articulated, it is crucial to integrate insights from prior loss-of-function studies. A comparison with existing research, such as investigations involving DNMT conditional knockout models, could provide valuable context and facilitate a more comprehensive understanding of the observed senescence and interferon responses.

> We understand your question and agree that it is important to integrate insights from prior loss-of-function studies. Broadly speaking, there are 3 such kinds of studies:

1-Genetic loss-of-function, constitutive or conditional.

Steve Baylin, Bert Vogelstein, and their colleagues, reported the first DNMT1 mutants in HCT116 cells (Rhee, Nature 2000; PMID 10801130). While the authors were aiming to generate a complete loss-of-function, it later emerged that their mutants are DNMT1 hypomorphs: they express a splicing variant of DNMT1, DNMT1 Δ E3–6, that is catalytically active and about 10% as abundant as DNMT1 in control cells (Egger PNAS 2006; PMID: 16963560; Spada JCB 2007; PMID: 17312023). The activity of the mutant is sufficient to sustain near-normal levels of DNA methylation (Rhee Nature 2000). We are not aware of any constitutive DNMT1 KO lines, presumably those fail to grow.

To circumvent the problems of constitutive KOs, En Li, Taiping Chen and their colleagues then developed an inducible DNMT1 KO, also in HCT116 cells (Chen Nat Genet 2007; PMID: 17322882). They found that Cre-mediated removal of the DNMT1 gene led to activation of the G2/M checkpoint and arrest in G2. Some cells escaped the checkpoint and either died during mitotic catastrophe or became tetraploid in G1. These phenotypes are vastly different from what we observe, and also inconsistent with RNAi (Spada JCB 2007; PMID: 17312023). Unfortunately, there are no rescue experiments showing that the phenotypes are due to the loss of DNMT1 itself, as opposed to an other phenomenon.

To the best of our knowledge, no constitutive UHRF1 KOs have been reported. An inducible KO was generated in HCT116 by Lei Li and colleagues (Tian Cell Reports 2015; PMID: 25818288). It is reported that "homozygous deletion of UHRF1 leads to severe proliferation defect (Figures S2B–S2D)", but the reason for the growth impairment is not investigated in the paper.

2-siRNA and shRNA studies

There is a myriad of papers in which knock-down of DNMT1 or UHRF1 was performed, in normal or cancer cells. They are not all consistent, which probably reflects the various cell types and experimental conditions used. We are not aware of any papers showing that

reducing UHRF1 and DNMT1 induces senescence in cancer cells. There are some indications that removing UHRF1 or DNMT1 from primary cells induces senescence (Jung JBC 2017 PMID: 28100769; Xiang Cell Discov 2017 PMID: 28626588; Xie Cell Death Dis 2017 PMID: 28277545), but the limitations of the experimental systems have restricted the mechanistic understanding.

3-degron approaches

One other paper has reported a degron approach on DNMT1 in cancer cells (Scelfo JCB 2024; PMID 38376). While this paper does not investigate senescence, it does report loss of proliferation, increased G1 population, and no increase in apoptosis, when DNMT1 is removed from DLD1 cells. All of these conclusions are consistent with our findings.

These considerations, and the papers cited above, are now included in the discussion section of the paper.

In Fig 1 depicting evaluations of senescence following DNMT1 and UHRF1 depletion, discrepancies are noted in the trends observed. For example, in Fig 1H, at Day 4 after depletion of DNMT1 alone shows no increase of SA-b-gal %area, while UHRF1 does, but in Fig 1F, at the same time DNMT1 depletion already caused nuclear area increase. There are other discordance between different panels of Fig 1. The authors should discuss about these differences.

> Thank you for helping us improve the rigor of the paper. You are absolutely correct that the different parameters we measured following DNMT1 and UHRF1 depletion do not vary in lockstep. As an example, as you rightly point out, nuclear area increases by day 4 of DNMT1 depletion, while SA-b-gal %area does not, while it is the opposite for UHRF1 depletion. We believe these discrepancies reflect two phenomena. First, not all the phenotypes are causally linked: increase of nuclear area and SA-b-gal %area can occur independently of each other. Second, while UHRF1 and DNMT1 have a shared function, the maintenance of DNA methylation, they also each have specific functions. As an example, we showed in our previous paper that UHRF1 controls de novo DNA methylation in HCT116 cells, while DNMT1 does not. Therefore, UHRF1 and DNMT1 may have different effects on senescence parameters at different time points. We agree it is important to discuss about these differences, and now do so in the "Discussion" part of the manuscript.

One noticeable gap in the current study is the limited experimental evaluation of DNA methylation changes, except for one LUMA assay on the global loss. To establish a more robust causal relationship between DNA methylation alterations and specific gene upregulation, a genome-wide analysis is deemed essential. Ideally, whole-genome bisulfite sequencing at selected time points for the degron lines or comparable conditions should be integrated with corresponding RNA-seq. Alternatively, leveraging related published datasets could serve as a surrogate to validate and elaborate on observed gene expression changes.

> Thank you for raising this key point. We agree that the limited experimental evaluation of DNA methylation changes was a gap of the initial study. Our previous publication (Yamaguchi 2024) supplied some of the data, specifically deep WGBS at days 0, 2, and 4. For this revision, we performed deep WGBS at days 6 and 8, and reanalyzed the whole set of data. Here is a summary of the new data and analyses provided in the paper and its revision:

	Day 0	Day 2	Day 4	Day 6	Day 8
RNA-seq	Yamaguchi 2024	Done for initial submission	Yamaguchi 2024	Done for initial submission	Done for initial submission
Deep WGBS	Yamaguchi 2024	Yamaguchi 2024	Yamaguchi 2024	Done for the revision	Done for the revision
Integrating WGBS and RNA-seq	Done for the revision				

The deep WGBS data is completely consistent with the data previously reported in Yamaguchi 2024 with LUMA, LC-MS, and shallow WGBS. Because the data are so similar, we have opted not to show them again as a figure in the manuscript. But here is a piece of data from Yamaguchi 2024 which you may find informative:

Reviewer panel: kinetics of DNA methylation loss, determined by WGBS

In addition, we followed your recommendation and integrated WGBS analysis with RNAseq analysis at the same time points. This is shown in the new panel 2G, copied below for your convenience. An interesting take-away from this analysis is that the EMT and p53 signatures we detect in bulk RNA-seq may be mere consequences of DNA demethylation at promoters. This seems less likely to be the case for the induction of the Interferon response, or for the SASP.

New panel 2G: integrated analysis of WGBS and RNA-seq data

Along this line, the manuscript would benefit from a deeper exploration of the mechanisms underlying the observed increase in p21 expression. Understanding the specific genomic regions affected by DNA methylation changes leading to p21 upregulation, or identifying the upstream triggers of this phenomenon, would enrich the interpretation of the findings and elucidate the molecular underpinnings of senescence induction.

> We thank you for stressing the importance of p21 regulation in the manuscript. Based on your suggestion we have:

-examined the WGBS data and found that methylation over the CDKN1A/p21 CpG island does not change over time (New Figure S4C).

-performed a cross-analysis of WGBS and RNA-seq data, which showed that p21 transcriptional induction was not correlated to DNA methylation levels at the promoter (New Figure S4D)

-and emphasized the data we previously presented in Figure S4. These data show that the senescent cells in our system downregulate 2 key repressors of p21: c-Myc and TFAP4. This is now summarize in the new schematic of Figure S4G, copied below.

New panels S4C-D: the induction of p21 is not caused by loss of DNA methylation

New panel S4G. The role of c-Myc and TFAP4 in p21 induction

Furthermore, the correlation between nuclear cGAS and SASP expression raises intriguing questions regarding the relationship between DNA hypomethylation and cGAS-mediated effects on senescence. Elaborating on the mechanisms linking DNA hypomethylation with the nuclear role of cGAS and its impact on target gene expression could provide new insights. Similar here, WGBS or methylation-sensitive PCR on regions respond to DNMT1/UHRF1 depletion would significantly enhance the manuscript's coherence and relevance to hypomethylation.

> Thank you for suggesting this. By analyzing WGBS and RNA-seq data we found that the cGAS promoter was losing DNA methylation during treatment of the cells (New panel S5A below), and that there was a strong correlation between loss of DNA methylation and RNA expression (New panel S5B below). In agreement with these data, it has been shown that the cGAS promoter is hypermethylated in CRC cells, including HCT116, and that its expression can be

induced by a 5-aza-dC treatment (Xia Cell Reports 2016 PMID: 26748708). Therefore, the simplest explanation for the cGAS induction we observe in our system is that it results from loss of DNA methylation over the cGAS promoter.

New panels S5A-B. Analysis of WGBS data at cGAS CpG island (A), correlation between promoter DNA methylation and expression (B).

Regarding the therapeutic implications highlighted in the study, I am not sure what to make of the discrepancy of Fig 5D and 5E, does that mean reduced level of senescence was not reflect in cellular growth? Please explain.

> Thank you for this perceptive observation, which indeed deserves an explanation. As you point out, removing cGAS from cells decreases the SA-b-gal positivity (Fig 5E) (and also SASP expression, as seen in Fig 5F), but it has no significant effect on cellular growth (Fig 5D). At this point, the simplest explanation is that cells without cGAS, even though they experience less senescence, grow slowly and/or apoptose more. Your point about this can affect therapeutic implications is well taken. However, unless we are mistaken, these results do not affect the logical conclusion that inhibiting UHRF1 or DNMT1, without causing DNA damage, may be an option to drive cancer cells into senescence. This conclusion is, of course, contingent on the existence of a therapeutic window where the same degree of inhibition is more deleterious to cancer cells than to normal cells.

Minor points:

At Page 8, one reference to Fig S1D should be to Fig S2D.

> Thank you for pointing out this mistake of ours. It has been corrected.

Fig 6E showed a much stronger reduction of Ki67 signal when UHRF1 is deprived than is DNMT1, but the quantification in Fig 6F showed the opposite. Why?

> Thank you for this perceptive observation. You are right, there was a disconnect between the image and the quantification. It just so happened that this particular image was not representative of the overall quantification (which is done on 7 fields). We have changed it.

Modified panel 6E. The old image for DNMT1^{AID2} was replaced with a new image that more accurately reflects what is seen on average, as shown in panel F.

Reviewer #4 (Remarks to the Author):

Chen et al. found that induction of global hypomethylation by the depletion of UHRF1 and/or DNMT1 proteins from HCT116 cells resulted in cellular senescence but not apoptosis. They further demonstrated that various types of cancer cells enter senescence upon the administration of a DNMT1 inhibitor. The authors' work represents the first report of hypomethylation-induced senescence in "cancer cells".

>Thank you for highlighting the novelty of our work.

Through well-designed and careful experiments, they demonstrated that this induced senescence is independent of p53 and Rb but involves p21 and cGAS. The authors also demonstrated that tumor cells can also be made senescent in vivo by decreasing DNA methylation.

>Thank you for this accurate summary. We are grateful for the time you took to examine our work, and glad that you found the experiments "well-designed and careful".

Although the authors made good use of their HCT116 cells with degron alleles to figure out which ones of known molecular pathways that underpin senescence are active in the demethylation-induced senescence of cancer cells, the weakness of this work is the lack of information through what mechanisms DNA hypomethylation trigger senescence in cancer cells. Many of the nonlethal stresses have been shown to induce cellular senescence as a stress response (PMID: 25312810), and DNA hypomethylation may be merely another example of the nonlethal stresses. To address this concern and improve the novelty and originality of this work, the authors need to clarify how DNA methylation directly prevents cancer cells from cellular senescence and how DNA hypomethylation triggers senescence pathways in cancer cells.

>Thank you for this perceptive point, and for directing us to the classic review published by the van Deursen team with the late Judith Campisi. We understand your concern that loss of DNA methylation could possibly be another nonlethal stress that triggers senescence, alongside well-known examples such as DNA damage, oxidative stress or ER stress.

Nevertheless, we think you will agree that a very large community of researchers, in the basic and medical/translational fields, works on DNA methylation. Therefore, providing a rigorous demonstration that DNA methylation triggers senescence, as we do, and a first characterization of the mechanisms that are involved (or not), is very relevant to many.

Major comments:

1. This reviewer thinks that the authors need to pursuit possible mechanisms through which DNA demethylation leads to senescence of cancer cells. In their previous work (Yamaguchi K et al. 2024, PMID: 38580649), the global DNA methylation level of HCT116 cells is shown to be dropped from 70% to 20-40% upon the depletion of DNMT1 and/or UHRF1.

Previous studies by other groups have demonstrated that such hypomethylation changes DNA replication timing and 3D genome organization in HCT116 cells (Qu Q et al, 2021, PMID: 34551299) and causes chromosomal instability possibly originated from centromeric hypomethylation in hTERT RPE1 cells (Besselink N et al, 2023, PMID: 37106015). Comparison of 3D genome organization in the authors' HCT116 cells with degron alleles upon DNA demethylation and that in replicative senescence may help narrow down genomic regions contributing to hypomethylation-induced senescence in cancer cells.

>Thank you for this insightful suggestion. We understand it could be interesting to perform Hi-C experiments on the degran cells at different time points to describe their 3D genome organization, and then compare it to published data, such as the organization of senescent cell nuclei published by Giacomo Cavalli's lab (Sati Mol Cell 2020, PMID: 32220303).

Nevertheless, among all the suggestions from 5 different reviewers, this is the one we have not followed up on. The reasons are both practical (cost, time, expertise needing for analyzing the data) and conceptual. Indeed, the single-cell RNAseq does show that the response is heterogeneous and we have a mixed population. Therefore, one would expect a mix of patterns in Hi-C that would be hard to deconvolute. Unless one moved to single-cell Hi-C, but that is an even taller order, and it feels out of proportion for the current revision work.

Again, we fully agree with you that changes in 3D genome organization may contribute to, or result from, senescence in our system, and this is something we mention in the discussion, but addressing this experimentally was beyond our capabilities for this revision, for which we already added 22 new result panels.

Because aneuploidy has been proposed as a key trigger into senescence (Macedo JC et al. 2018, PMID: 30026603; Kirsch-Volders M, Fenech M. 2023, PMID: 37866738), it is important to determine the frequency of aneuploidy upon induction of hypomethylation in the authors' HCT116 derivative cells.

> This is an important point, thank you for bringing it up. As you say, aneuploidy has been proposed as a key trigger into senescence, so it is important to determine if this phenomenon is at play in the senescence we observe upon loss of DNA methylation.

To answer your question, we leveraged the scRNA-seq generated for the revision, and inferred CNVs from the expression data. The results are shown in the new panel below.

New panel: Analysis of scRNA-seq does not suggest widespread aneuploidy (a more compact version of this figure, without the diploid controls, is new panel Fig.S3M)

We used InferCNV, and validated our analysis by using the reference data provided in the package (control diploid and aneuploid cells). We observe in our HCT116 cells the local gains and losses of genomic DNA that have been reported for HCT116 cells by gDNA sequencing (Cohen-Sharir et al, Nature 2021, PMID: 33505028). Furthermore, the analysis shows no difference between the degran and control HCT116 cells, arguing against an implication of aneuploidy in our system.

Two pieces of published data also support this conclusion:

1) senescence induced by aneuploidy is p53-dependent in HCT116 cells (Giam et al Cell Cycle 2020, PMID 33305692; Narkar Cell Reports 2021, PMID 33761356). We show that senescence induced by loss of DNA methylation does not require p53, therefore it is unlikely to result from aneuploidy.

2) DLD1 cells with a DNMT1 degron allele do not show increased aneuploidy as DNA methylation decreases (Scelfo JCB 2024; PMID 38376465, Figure S4E).

Induction of genome-wide demethylation in cancer cells may lead to reactivation of retrotransposable elements and loss of heterochromatin, the former of which has been shown to drive IFN response in senescent fibroblasts (De Cecco M et al, 2019, PMID: 30728521) and the latter of which has been shown to be implicated in replicative senescence (Zhang X et al. 2021, PMID: 34140314; Deng L et al. 2019, PMID: 31350386). By examining the extent of each of the events listed here in the authors' HCT116 derivative cells upon the induction of demethylation, the authors will be able to narrow down possible mechanisms of demethylation induced senescence in cancer cells.

> Thank you for these insightful suggestions. We have now examined in more details the two events you mentioned: reactivation of retrotransposable elements and loss of heterochromatin. Here are the observations we made:

A-Reactivation of retrotransposable elements.

We focused on L1 elements, as they are shown to drive the IFN response in the paper you point out. We teamed up with an L1 specialist, Dr Gael Cristofari, to ensure we did everything correctly. Dr Cristofari and his coworker Sophie Lanciano have therefore been added to the author list in the revised paper. With their help, we investigated two sub-questions:

1-Are retrotransposable elements induced upon induction of demethylation?

We performed western blotting with a thoroughly validated antibody directed against ORF1p of L1. As a positive control of L1 induction, we treated HCT116 cells with 5-aza-cytidine (1 micromolar for 5 days). This experiment is now presented as the new panel S3H, copied below for your convenience.

New panel S3H: expression of L1 ORF1p

We observe an induction of ORF1p, at the later time points, and most prominently in the double degron line (UHRF1-AID/DNMT1-AID). This led us to our following question.

2-Are retrotransposable elements involved in senescence in our system?

For this we performed shRNA against L1, using a vector published by Dr Cristofari (Philippe ELife 2016, PMID: 27016617). We infected and selected the degron cells, then treated them with 5-aza-C to induce ORF1p to monitor the efficiency of the shRNA. This experiment now appears as panel S3I in the paper, copied below for your convenience:

New panel S3I: validation of shORF1 vector

The conclusion from this experiment is that the shRNA vectors works, as previously published (Philippe ELife 2016). Next, we asked whether stably knocking down L1s had an effect on proliferation or senescence in our cells. As shown in the new panel S3J, copied below, the shORF1 vector did not rescue the growth defect in any of our cell lines.

New panel S3J: no rescue of growth defect by shORF1 vector

Besides the growth phenotype, we also assessed whether shORF1 remediated SA-beta-gal staining or SASP expression, and found that it did not (New panels S3K-L, below).

New panels S3K-L: no rescue of SA-beta-gal staining or SASP expression by shORF1 vector

B-Loss of heterochromatin. We looked at a number of heterochromatin marks by western blotting. We selected day 4, as it is an intermediary time point where we are more likely to observe effects that are causal to senescence, and less likely to observe effects that are just a consequence of senescence. In addition, doing the experiment at day 4 decreases the confounding effect of increased nuclear size. As you can observe in the figure below, there is no loss of H3K9me2 or H3K27me3 in the degron cells, even though they are in the process of senescing. As for H3K9me3 there is a mild decrease, specifically in the U^{DAID1} cells. In contrast, there is a marked increase of H3K9ac in U^{AID1} and U^{DAID1} cells. This is an interesting finding that we now comment upon.

New panel S2J. Global chromatin analysis at day 4 of treatment.

The promoters of cGAS and STING are reported to be highly methylated and their expression is repressed in cancer cells (PMID: 29367762). Rather than genome-wide changes triggered by DNA demethylation, can the demethylation-induced senescence in cancer cells be explained by the de-repression of a limited number of such senescent pathway genes? Such a possibility should also be assessed and discussed in this paper.

> Thank you for pointing out this interesting reference by the Barber lab. Another excellent paper by the same team predates it and also explores the role of DNA methylation on cGAS and STING expression (Xia Cell Reports 2016, PMID: 26748708), it is the one we cite in our paper. Thanks also for your important suggestions. To address them, we carried out the following analyses and experiments. Please note that we left STING aside, as it is not involved in the senescent phenotype.

1-Analysis of WGBS data

We generated deep WGBS data at all experimental time points, analyzed that set, and correlated it with RNAseq. As shown in new Figure S5A, the promoter of cGAS loses methylation upon loss of DNMT1 and UHRF1. In addition, the induction of cGAS transcript in RNA-seq correlates with loss of DNA methylation at the promoter (new Figure S5B). These results are consistent with a model in which it is the loss of DNA methylation that triggers cGAS expression.

New panels S5A-B. Analysis of WGBS data at cGAS CpG island (A), correlation between promoter DNA methylation and expression (B).

2-Overexpression of cGAS

Our data show an induction of cGAS during senescence. As you rightly propose, it could be that the increased expression of cGAS following loss of DNA methylation is in itself sufficient to cause senescence. We tested if this was the case by forced expression of cGAS in HCT116 cells, using either a WT form, or a constitutively nuclear form (NLS-cGAS). Both proteins were expressed and localized as expected, yet neither drove the cells into senescence (new figures S5D and S5E, copied below).

New panels S5D-E. Forced expression of cGAS or NLS-cGAS does not induce senescence

2. The following two papers are highly relevant to the authors' present work and should be appropriately cited and discussed in the authors' manuscript. Xie et al. (2017, PMID: 28277545) demonstrated that DNMT1 knockdown in young human skin fibroblasts induced senescence phenotype. Young et al. (2017, PMID: 28394343) demonstrated that UHRF1 knockdown in human fibroblasts (2BC and IMR90 lines) induced senescence.

> Thank you for pointing out us towards these relevant papers. They are now mentioned in the introduction. We have also added Xiang et al (Cell Discov 2017, PMID: 28626588), another good paper that came out the same year and comes to similar conclusions in a different system.

Reviewer #5 (Remarks to the Author):

> We thank you for the time you have taken to review our work.

REVIEWERS' COMMENTS

Reviewer #1 (Remarks to the Author):

The authors have made major efforts to address the comments of the four reviewers. They added 22 new figure panels to the already existing one hundred or so. The new data have significantly improved the manuscript. I am satisfied with the revisions.

Reviewer #2 (Remarks to the Author):

The authors have addressed my questions satisfactorily. I have no further comments.

Reviewer #3 (Remarks to the Author):

The authors have fully addressed my concerns. The revised manuscript is significantly improved with the addition of substantial new data and analyses that rule out alternatives, and strengthen the main conclusions. While the ultimate upstream molecular sensor that links DNA hypomethylation to the downstream effectors (like p21 and cGAS activation) remains to be identified, the current study provides a thorough characterization of this novel, non-canonical senescence pathway in cancer cells and its key features.

Reviewer #4 (Remarks to the Author):

The authors satisfactorily addressed my comments. I found the reasons the authors mentioned as to why they did not examine changes in 3D genomic organization were reasonable. I hope the authors clarify the causal mechanisms of induced senescence by DNA demethylation in cancer cells through additional efforts such as genetic screens and thorough epigenetic profiling in the future.

Reviewer #5 (Remarks to the Author):

> We thank the five reviewers for accepting the papers without further changes